# Quality Not Quantity: On the Interaction between Dataset Design and Robustness of CLIP

**Thao Nguyen**
University of Washington
thaottn@cs.washington.edu

**Gabriel Ilharco**
University of Washington
gamaga@cs.washington.edu

**Mitchell Wortsman**
University of Washington
mitchnw@cs.washington.edu

**Sewoong Oh**
University of Washington
sewoong@cs.washington.edu

**Ludwig Schmidt**
University of Washington,
Allen Institute for Artificial Intelligence
schmidt@cs.washington.edu

## Abstract

Web-crawled datasets have enabled remarkable generalization capabilities in recent image-text models such as CLIP (Contrastive Language-Image pre-training) or Flamingo, but little is known about the dataset creation processes. In this work, we introduce a testbed of six publicly available data sources—YFCC, LAION, Conceptual Captions, WIT, RedCaps, Shutterstock—to investigate how pre-training distributions induce robustness in CLIP. We find that the performance of the pre-training data varies substantially across distribution shifts, with no single data source dominating. Moreover, we systematically study the interactions between these data sources and find that combining multiple sources does not necessarily yield better models, but rather dilutes the robustness of the best individual data source. We complement our empirical findings with theoretical insights from a simple setting, where combining the training data also results in diluted robustness. In addition, our theoretical model provides a candidate explanation for the success of the CLIP-based data filtering technique recently employed in the LAION dataset. Overall our results demonstrate that simply gathering a large amount of data from the web is not the most effective way to build a pre-training dataset for robust generalization, necessitating further study into dataset design. Code is available at https://github.com/mlfoundations/clip_quality_not_quantity.

## 1 Introduction

Large models pre-trained on web-scale datasets are becoming a cornerstone of machine learning. For instance, the past two years have witnessed the arrival of several new models such as GPT-3 [12], Chinchilla [33], and PaLM [15] for natural language processing, or CLIP [48], BASIC [46], and Flamingo [3] for computer vision. These models exhibit unprecedented generalization capabilities in zero-shot inference, in-context learning, and robustness to distribution shift. A key ingredient enabling their generalization performance are the large and diverse pre-training corpora that exceed previous datasets by multiple orders of magnitude. For instance, the training set of BASIC [46] contains 6.6 billion images, which is more than 1,000 times larger than the widely used ImageNet [18] training set from the 2012 competition containing 1.2 million images.

Despite the central role datasets play for pre-trained models, little is known about them, especially for image-text models. The aforementioned CLIP [48], BASIC [46], and Flamingo [3] all rely on datasets internal to the respective organizations, which is also the case for other models such as DALL-E [49], Florence [63], and ALIGN [35]. In addition, research publications often provide little details on the data collection processes, e.g., the data sources or data filtering mechanisms. Beyond clear issues

36th Conference on Neural Information Processing Systems (NeurIPS 2022).

such as reproducibility and the potential presence of harmful content, the opaque dataset creation practices also make it hard to identify effective methods for assembling pre-training datasets. As a result, researchers cannot build on each other's dataset innovations, which obstructs the incremental research process that has successfully accumulated algorithm and architecture improvements in machine learning models. A more principled understanding of dataset creation will likely enable further progress in the generalization capabilities of pre-trained models.

A basic approach to dataset creation would be to simply train on *all* available data of a given type. While scaling up training sets has indeed been integral to the recent progress in large models, advances in weak and self-supervision [14, 29, 30, 12, 20] have led to an abundance of potential training data. For instance, the large LAION-5B dataset [2] of 5 billion image-text pairs is itself a subset of about 50 billion images from Common Crawl. This abundance is already exceeding the amount of data models can currently be trained on within a reasonable time,[1] making the aforementioned baseline of using all available data infeasible. Consequently, deciding *what* data to train on is becoming increasingly important.

In this paper, we take a step towards a better understanding of pre-training data and investigate the impact of dataset design on the generalization capabilities of image-text models. Specifically, we focus on CLIP, the first large model of this kind that demonstrated remarkable robustness to multiple challenging distribution shifts. The main questions we ask are: (*i*) How much do different web data sources vary in their induced robustness? (*ii*) Do dataset combinations lead to better robustness? (*iii*) Can filtering with an existing image-text model improve data quality? We address these questions from both an experimental and theoretical perspective.

On the experimental side, we begin by assembling a corpus of six different datasets from the web, spanning a variety of sources including Flickr, Shutterstock, Wikipedia, Common Crawl, and Reddit. We then measure the robustness of CLIP models trained on each dataset. In particular, we compare the zero-shot accuracy of these models on ImageNet and a set of canonical ImageNet distribution shifts. We find that the robustness induced by each pre-training dataset varies widely, and that sources with careful curation such as Wikipedia do not necessarily outperform those with minimal filtering.

In addition, the datasets cannot be compared along a single dimension: different datasets help with robustness to different distribution shifts. This in turn motivates studying how combining datasets affects robustness. We find that a model trained on two datasets does not inherit the robustness properties of both. Rather, while the model is exposed to both distributions, its robustness interpolates between that of the individual datasets. This indicates that dataset designers must carefully combine the pre-training data in order to preserve and enhance the robustness of the resulting models.

Building on our empirical results, we introduce a theoretical model to better understand our experimental findings. Our model is simple enough to allow for mathematical analysis and can still capture some phenomena of real-world, web-crawled datasets. Specifically, our theoretical model also shows that combining multiple datasets dilutes the robustness of the better data distribution. Moreover, our model provides a candidate explanation for another interesting phenomenon in the curation of pre-training data: the LAION-400M experiment [52] and follow-up CLIP reproductions [2] demonstrated that filtering a noisy source (Common Crawl) with a CLIP model results in a dataset on which newly trained CLIP models exhibit *higher* robustness to some distribution shifts.

**Paper outline.** We first briefly review related work and the relevant background on robustness to distribution shift (Section 2), before discussing our experimental setup in Section 3. Sections 4 and 5 then measure the robustness of CLIP induced by individual data sources and their combinations. To support these empirical results, Section 6 presents our theoretical analysis. We conclude with future research directions in Section 7.

## 2  Background & Related Work

**Vision-language models.**   Large vision-language models like CLIP and ALIGN have become an active area of research owing to their success on various computer vision tasks [48, 35]. Existing work expands their capabilities by either increasing model and dataset size [46], or by using additional supervision as in DeCLIP [40], SLIP [44], and FILIP [62]. In contrast, we study the effect of *pre-training data composition* on task performance with a focus on robustness.

---

[1]Recall that the training set for Google's largest image-text model ALIGN contains "only" 6.6 billion images, making it about eight times smaller than the source dataset for LAION-5B.

In a recent work, Fang et al. [22] showed that CLIP's robustness primarily stems from its diverse pre-training distribution and not training set size, language supervision, or contrastive loss functions. However, Fang et al. [22] only conducted experiments with two datasets (YFCC-15M and ImageNet-Captions), one of which contains less than one million images. We take the insights of Fang et al. [22] as our starting point and expand the range of pre-training datasets to six different sources, each containing at least five million images. This enables us to study the robustness induced by various pre-training sources, and how dataset combinations affect robustness.

**Distribution shift.** Robustness to distribution shift is a long-standing issue in machine learning [58, 47] and has recently received renewed attention as researchers scrutinize the generalization performance of neural networks in greater detail [55, 8, 9, 5, 51, 27, 16, 38]. Similar to CLIP [48], we focus on robustness to natural distribution shifts, where the corresponding test sets contain only unmodified images and are not intentionally perturbed by adversarial examples or synthetic corruptions (such as Gaussian noise or blur patterns) [31]. Specifically, we test robustness on ImageNetV2 [50], ImageNet-R [32], ImageNet-Sketch [59], and ObjectNet [6] because prior work has established many baselines for these distribution shifts [56]. Moreover, models robust to these shifts also show improved robustness on other out-of-distribution benchmarks such as WILDS [36, 60].

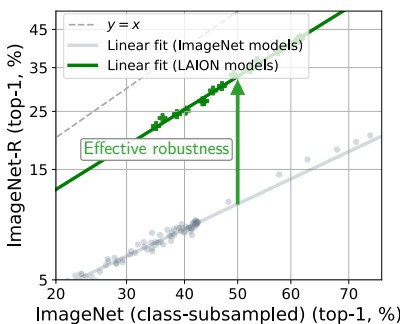

Figure 1: Models pre-trained on LAION exhibit *effective robustness* [56] compared to standard models trained on ImageNet. Effective robustness is defined as movement towards a classifier which is robust to distribution shift. A classifier is more robust the closer it is to the $y = x$ line. A classifier on the $y = x$ line is not affected by the distribution shift.

The robustness literature usually discusses distribution shift in terms of "in-distribution" and "out-of-distribution" data. This terminology is natural when data from the same distribution as the "in-distribution" test set is used for training or fine-tuning. However, it is unclear what counts as "in-distribution" when models are trained on large-scale, generic pre-training datasets that aim to improve performance on a wide variety of tasks. To address this issue, we follow the more flexible definition of distribution shift employed in prior work [56, 43] and measure robustness as accuracy difference between two related but distinct test distributions (e.g., ImageNet and ImageNet-Sketch). The expectation is that the shift between the two test distributions should not affect an ideal robust model, for instance because the shift does not affect the accuracy of humans labelers [53]. Taori et al. [56] defined this notion of robustness as *effective robustness*. Radford et al. [48] adopted the effective robustness framework in the evaluations of their CLIP model. Effective robustness is illustrated in Figure 1 and measures movement towards a perfectly robust classifier which is not affected by the shift between two test distributions.

Prior work [43] has evaluated several hundred models and demonstrated experimentally that changes to model architecture, training set size, training algorithm, and other model-related factors do *not* change effective robustness in most cases. In contrast, changes to the pre-training *distribution* can improve the effective robustness of a model. This makes effective robustness a useful metric for evaluating the influence of pre-training data sources on robust generalization, since it removes confounders stemming from model hyperparameters and the number of training samples. The effect of the pre-training distribution on effective robustness is particularly pronounced when models are evaluated in a zero-shot setting: while training on a target distribution can improve accuracy on that distribution [60, 25, 64, 65, 61], this process can deteriorate robustness to distribution shift [48, 60, 4, 46].

The existence of a universal linear trend for accuracies on a pair of test sets has been analytically studied in [43]. For a simple binary classification model similar to our Assumption 1 (see Section 6), convergence to a linear trend is shown with the deviation from the line scaling as $O(1/\sqrt{d})$. However, the analysis in [43] is restricted to a specific stochastic distribution shift for the out-of-distribution test data and a fixed classifier independent of the training data. Hence the dependence of the slope on the training set size, variations in the training methods, and properties of the training distribution cannot be described in their model. We provide significantly more fine-grained analyses (Theorem 1) that captures all such tradeoffs, which allows us to draw novel insights into how data mixing (Theorem 2) and filtering (Theorem 3) affect model robustness.

## 3  Experiment Setup

**Model.** We focus on the CLIP model [48] which has demonstrated unprecedented zero-shot performance on a wide range of downstream tasks, as well as robustness to various distribution shifts [43]. Given an image-text pair, CLIP is trained to maximize the cosine similarity between the embeddings of the text and that of the image, relative to the similarity of unconnected image-text pairs. We use the CLIP implementation from the OpenCLIP GitHub repository [34], with ResNet-50 [28] as the image encoder model. We vary the pre-training set size and hyperparameters such as number of epochs to obtain different accuracies on each data distribution. Due to compute constraints, the total training set size is at most 15M samples for most of our experiments. Appendix B contains more training details.

**Data.** To study the effects of training distributions on robust generalization, we collect several datasets from publicly available sources. Most of these have been studied in the context of various vision and language tasks in previous work:

- YFCC: We experiment with the 15M subset of the YFCC100M dataset [57] that the original CLIP paper [48] used for dataset ablation studies. The images and captions are collected from Flickr.
- LAION [52]: The images and corresponding alt-texts come from web pages collected by Common Crawl [1] between 2014 and 2021. We randomly select a subset of 15M samples to experiment with, and ensure that the accompanying NSFW tags of all chosen images are 'UNLIKELY'.
- Conceptual Captions [13]: We use CC-12M for our experiments, which consists of images and HTML alt-text from an unspecified set of web pages.
- RedCaps [19]: This dataset contains 12M examples, obtained from 350 manually curated subreddits between 2008 and 2020. The subreddits are selected to contain a large number of image posts that are mostly photographs and not images of people.
- Shutterstock: 15M images and captions were crawled from the Shutterstock website in 2021.
- WIT [54]: Image-text pairs come from Wikipedia pages. We use reference description as the source of text data and obtain 5M examples after filtering to include only English language examples.

Appendix A.1 contains an analysis of image and text statistics, as well as randomly selected data samples from each source.

**Evaluation.** We pick ImageNet as the reference distribution and evaluate CLIP on four natural distribution shifts derived from ImageNet: (i) ImageNet-V2 [50]—a reproduction of ImageNet test set using the original dataset creation process, (ii) ImageNet-R [32]—renditions (e.g., sculptures, paintings) for 200 ImageNet classes, (iii) ImageNet-Sketch [59]—sketches of ImageNet class objects, and (iv) ObjectNet [6]—a test set of objects in novel backgrounds, rotations and viewpoints with 113 classes overlapping with ImageNet. Visualizations of random samples from each distribution shift can be found in Appendix A.2.

## 4  Individual Pre-training Data Sources

We first investigate how well a CLIP model trained on each data source would perform under different distribution shifts of interest. As seen from Figure 2, while all sources yield the same linear trend on ImageNet-V2, some display clear advantages when evaluated on other test distributions. For example, Shutterstock offers the best out-of-distribution performance on ImageNet-Sketch, while RedCaps displays the most effective robustness on ObjectNet. On ImageNet-R, LAION, CC12M and Shutterstock seem to do much better than the rest. Overall no pre-training data distribution is consistently the most robust across all evaluation settings.

We also measure the data efficiency of each source, i.e., how much the performance would change with more samples from the same source, see Appendix C. Similar to the previous observation, the six pre-training sources display vastly different data efficiency depending on the distribution shifts of interest. Although LAION and CC-12M exhibit similar effective robustness in all evaluation settings in Figure 2, this analysis reveals subtle differences between these two training distributions in the low-data regimes.

## 5  Combining Data Sources

In the previous section, we demonstrate the variability in behavior of different data sources based on the distribution shift at test time. A natural question then arises from this observation: does combining

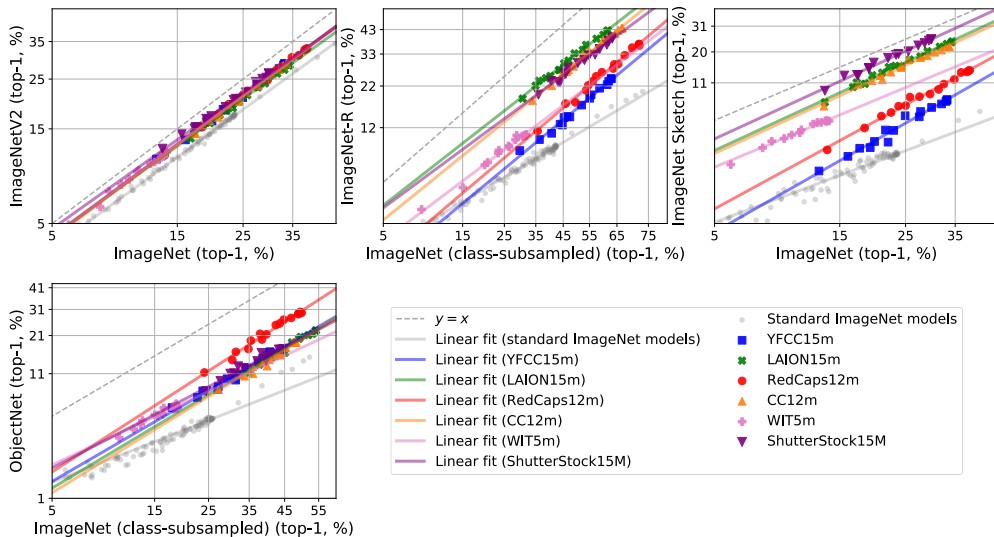

Figure 2: **Performance of the six pre-training data sources under various distribution shifts.** We find that the behavior—both in terms of accuracy and the slope of the linear trend—of the pre-training data varies substantially across distribution shifts, with no single data source dominating. Most shifts help highlight the strengths and weaknesses of different data sources, except for ImageNet-V2, where the linear trends produced by individual sources are highly correlated with one another.

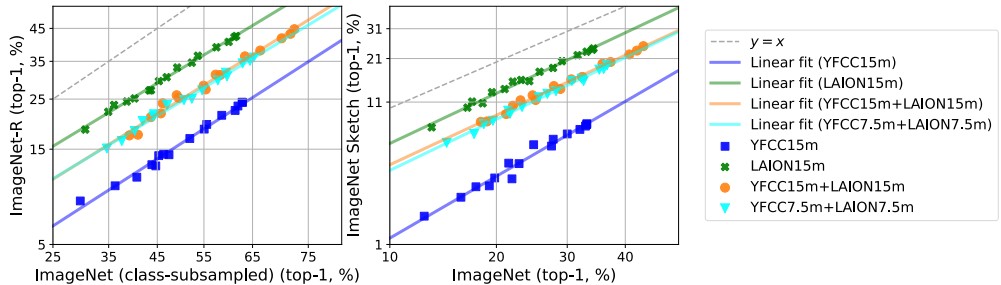

Figure 3: **Combining YFCC and LAION training data in equal ratios produces models with intermediate robustness.** Given a fixed data budget of 15M samples, the linear trend produced by training CLIP on a YFCC-LAION data mixture, with 7.5M datapoints from each source (cyan line), lies between that of training CLIP on YFCC (blue line) and LAION (green line) entirely. Even when we increase the total training set size (30M) and use all data available from both sources (orange line), the same pattern persists.

multiple sources help improve robustness across some, if not all, test distributions of interest? We investigate two common approaches of aggregating information from different training sets—input mixing and output ensembling. In the subsequent discussion, we focus on distribution shifts that bring out significant differences in behavior across the pre-training data sources (e.g., ImageNet-R and ImageNet-Sketch); the full results on all distribution shifts can be found in Appendices D and E.

## 5.1 Input Mixing

In input mixing, we randomly select and combine samples from multiple data sources to build the pre-training dataset. Our findings indicate that this exposure to more training distributions doesn't help CLIP take advantage of the complimentary strengths of each source. Rather, the effective robustness of the resulting model is less than that of training on the best individual source for each distribution shift.

We start with combining data from two largest data sources in the testbed—YFCC and LAION. To remove dataset size as a confounder, we fix the total amount of training data at 15M samples, and sample 7.5M image-text pairs from each source. This data mixture yields a linear trend that lies in between the trends obtained from training on 15M YFCC and 15M LAION datapoints separately

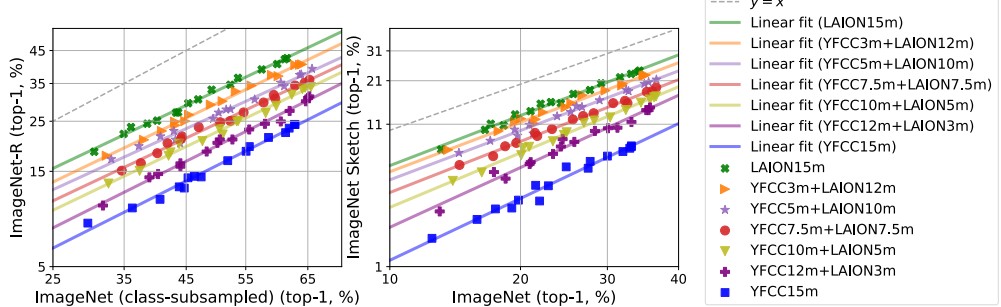

Figure 4: **Varying the sample contributions of YFCC and LAION to the input data mixture produces a smooth interpolation of the linear trend between those of training on YFCC and LAION separately.** Keeping the total number of training samples fixed at 15M, as we vary the contribution of YFCC to the dataset mixture from 15M (i.e., only training on YFCC) to 0M (i.e., only training on LAION), the resulting linear trend gradually shifts from that of YFCC-15M (blue line) to that of LAION-15M (green line).

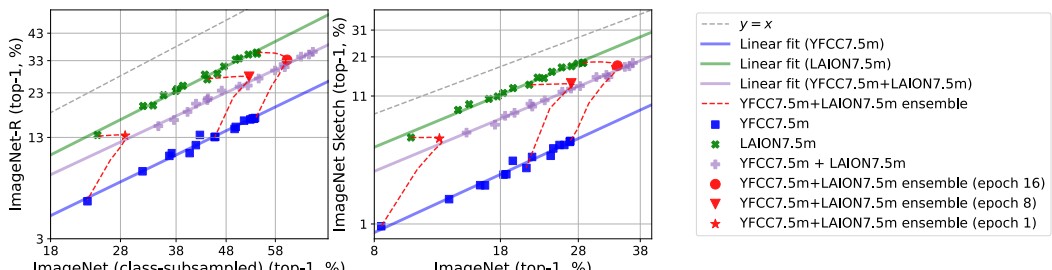

Figure 5: **Ensemble outputs of CLIP models trained on YFCC and LAION separately share the same linear trend as a single model trained on the combined data mixture (where each source contributed equally).** We ensemble the logit predictions of YFCC-trained (blue line) and LAION-trained (green line) models taken from the same epoch, with varying ensemble weights between 0 and 1 (red dashed line). When the outputs are combined with equal weights (red markers), the resulting test accuracies closely track the linear trend produced by pre-training CLIP on a data mixture with equal number of samples from each source (purple line).

(Figure 3). Even when we remove the constraint on the training set size and use all the data available from these sources (i.e., 30M samples), the same observation on robustness holds, and the resulting linear trend is highly correlated with that of training on the YFCC-7.5M + LAION-7.5M mixture.

The previous set of experiments combines YFCC and LAION data with a 50:50 ratio. In Figure 4, we find that when this ratio is varied within a fixed budget of 15M datapoints, the robustness linear trends of the corresponding mixtures form a smooth interpolation between the linear trends of training on 15M YFCC and 15M LAION samples separately. Experiments with different combinations of sources, as well as mixtures of a larger number of sources, can be found in Appendix D. There we also include results for combining data from CIFAR-10 and CINIC-10 distributions to train ResNets on image classification tasks, where we observe the same interpolation pattern. We provide a theoretical justification for this phenomenon in Section 6.2.

## 5.2 Output Mixing

Another common approach to take advantage of different training sets is to combine them at model output level (i.e., ensemble) [21, 39, 26, 23, 7, 11, 24, 45]. Here, we train a CLIP model on each pre-training distribution of interest and combine the logit predictions of all the resulting models with equal weights. In experiments with mixing YFCC and LAION (Figure 5), the ensembles of 2 single-source-trained CLIP models taken from the same epoch of training, lie on the linear trend of a *single* CLIP model trained on the combined data. This holds across different distribution shifts that we consider. The same observation also applies when we ensemble 6 CLIP models trained separately on the 6 data sources collected, with the contribution of each model being weighted equally. Refer to Appendix E for more details.

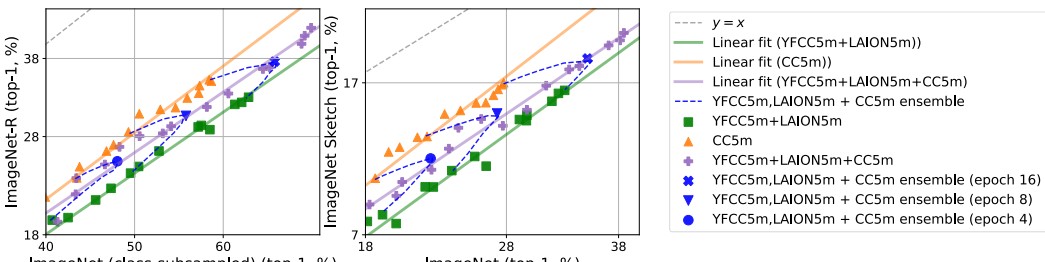

Figure 6: **Using ensemble outputs to predict the linear trend of input mixing without retraining CLIP from scratch.** A generalization of the observation made in Figure 5 is that given an existing pre-training dataset that could be a mixture (e.g., YFCC-5M + LAION-5M, green line) and a new data source (e.g., CC-5M, orange line), we could use the ensemble outputs (blue markers) of two CLIP models that have been trained separately on these two data distributions, to estimate where the linear trend for a CLIP model trained on *all* the data would lie (purple line). This removes the need to actually train CLIP from scratch on the now bigger 3-source mixture.

We next show that this phenomenon extends to more complex mixtures. The fact that output mixing (i.e., ensembling the predictions of CLIPs trained on individual sources) is predictive of the linear trend produced by input mixing (i.e., training on *all* the data from these sources) presents an opportunity to estimate model robustness given new data sources, without having to train the model from scratch on the new, potentially much larger, combined dataset. For example, in Figure 6, assuming access to a CLIP model trained on the YFCC-5M + LAION-5M mixture, and another one trained on CC-5M, the ensembles of these two models across different stages of training share the same linear trend as a CLIP model trained on the YFCC-5M + LAION-5M + CC-5M mixture. In Appendix E, we show the results of ensembling 6 CLIP models trained with the 6 data sources we collected, as well as the CINIC-10 + CIFAR-10 ensemble, a uni-modal image classification setting where output mixing accuracies are also predictive of the linear trend of input mixing.

## 6 Analysis under Simple Binary Classification Models

Empirically we observe, for e.g., in Figure 2, that for a pair of test datasets $(\mathcal{D}_1, \mathcal{D}_2)$, the corresponding test accuracies after probit transform achieved by various trained models (including different architectures and training algorithms) on the same training dataset $\mathcal{D}$ all lie on the same line. Furthermore, this line includes models trained on only a subset of $\mathcal{D}$ [43]. In other words, there is a universal line that is determined only by the training data distribution and the two test distributions, and is independent of which architecture we use, how we train the model, or how many samples we use from the training set. To explain this phenomenon, we study the following class of trained models parametrized by $(\theta \in \mathbb{R}^d, \rho \in \mathbb{R}_+)$ representing the training distribution, with $n$ representing the training data size, and $\xi \in \mathbb{R}_+$ representing the variations due to the training algorithm.

### 6.1 Universality of Accuracy on the Line for Binary Classification

We analyze a simple but canonical binary classification example where each training data is parametrized by its ground-truth linear classifier $\theta \in \mathbb{R}^d$ and its Signal-to-Noise Ratio (SNR) $\rho^2 \in \mathbb{R}_+$. Concretely, a training dataset $\mathcal{D}_{n,\theta,\rho} = \{(x_i \in \mathbb{R}^d, y_i \in \pm 1)\}_{i=1}^n$ is a set of $n$ i.i.d. paired samples from a joint distribution $P_{\theta,\rho}$ defined as follows.

**Assumption 1** (Data distribution). *We define a joint distribution $(x_i, y_i) \sim P_{\theta,\rho}$ as (i) $y_i = \pm 1$ uniformly at random, and (ii) $x_i = y_i\theta + (\|\theta\|/\rho)z_i$ where the noise $z_i$ is zero-mean and has independent entries with variance one. For an observation $(x_i, y_i)$, we refer to $\|\theta\|^2$ as the signal power and $\|\theta\|^2/\rho^2$ as the corresponding noise power with SNR $\rho^2$.*

**Assumption 2** (Trained model distribution). *We consider a (random) linear model parameter $\hat{\theta}_{n,\xi} \in \mathbb{R}^d$ that predicts a binary label $\text{sign}(\langle x, \hat{\theta}_{n,\xi} \rangle)$ for a test example $x \in \mathbb{R}^d$. We assume that the random model $\hat{\theta}_{n,\xi} = \theta + (\xi\|\theta\|/(\rho\sqrt{n}))z \in \mathbb{R}^d$, trained on $\mathcal{D}_{n,\theta,\rho}$ is unbiased, $\mathbb{E}[\hat{\theta}_{n,\xi}] = \theta$, and that $z$ has independent entries with variance one each. The randomness comes from the training data as well as any internal randomness in the training algorithm. The parameter $\xi \in \mathbb{R}_+$ captures the variations in the resulting model distribution due to changes the training algorithm.*

Concretely, one canonical example of a trained model is $\hat{\theta}_{n,\xi} = (1/n)\sum_{i=1}^{n} y_i x_i$. It follows that $\hat{\theta}_n = \theta + (\|\theta\|/(\rho\sqrt{n}))z$ with $\xi = 1$. Hence, $\xi$ measures the randomness of the trained model relative to this simple training algorithm.

We analyze the resulting accuracy when evaluated on two test distributions $P_{\theta_1,\rho_1}$ and $P_{\theta_2,\rho_2}$ as defined above, with $\mathrm{Acc}_{\theta_1,\rho_1} := P_{\theta_1,\rho_1}\{\mathrm{sign}(\langle X, \hat{\theta}_{n,\xi}\rangle) = Y\}$, and $\mathrm{Acc}_{\theta_2,\rho_2}$ defined similarly. In particular, we are interested in how the accuracy pair $(\Phi^{-1}(\mathrm{Acc}_{\theta_1,\rho_1}), \Phi^{-1}(\mathrm{Acc}_{\theta_2,\rho_2}))$ behaves as we vary the sample size $n$ and as we vary the training algorithm represented by $\xi$. Here, $\Phi^{-1} : [0,1] \to \mathbb{R}$ is the inverse of the CDF of a standard Gaussian distribution, defined as $\Phi(t) = \mathbb{P}(z \leq t)$ where $z \sim \mathcal{N}(0,1)$. $\Phi^{-1}$ is also called the probit function. This choice of mapping the accuracy with the probit function is critical in getting the linear relation, which we will explain in Remark 1.

**Theorem 1** (Universality of accuracy on the line). *Under Assumptions 1 and 2, asymptotically as the dimension $d$ grows linearly in the sample size $n$ such that $\lim_{d\to\infty} n/d = \alpha^2$, we have*

$$\lim_{d\to\infty} \mathrm{Acc}_{\theta_1,\rho_1} = \Phi\left(\cos(\theta_1,\theta)\frac{\rho_1\rho\alpha}{\xi}\right), \text{ and } \lim_{d\to\infty} \mathrm{Acc}_{\theta_2,\rho_2} = \Phi\left(\cos(\theta_2,\theta)\frac{\rho_2\rho\alpha}{\xi}\right). \quad (1)$$

*Further, under Assumption 5 in the Appendix, for some universal constant $c > 0$,*

$$\Phi^{-1}(\mathrm{Acc}_{\theta_2,\rho_2}) = \frac{\cos(\theta_2,\theta)\rho_2}{\cos(\theta_1,\theta)\rho_1}\Phi^{-1}(\mathrm{Acc}_{\theta_1,\rho_1}) + O\left(\frac{e^{\frac{cn}{d}}}{\sqrt{n}}\right). \quad (2)$$

We provide a proof in Appendix F.1. This analysis implies that for any training sample size $n$ and any (variation due to the) training algorithm $\xi$, the resulting accuracy pair after probit transform lies on a *universal* line determined by Eq. (2), and the slope only depends on the two test distributions $(P_{\theta_1,\rho_1}, P_{\theta_2,\rho_2})$ and the training distribution $P_{\theta,\rho}$. The similarity between the training data and each test dataset is captured by the angles: $\cos(\theta_1,\theta)$ and $\cos(\theta_2,\theta)$. More similar training data achieves a larger test accuracy. The hardness of the test distribution is captured by the SNR of each test dataset: $\rho_1$ and $\rho_2$. We emphasize that this line is universal in the sense that the slope does not depend on the sample size $n$ and training algorithm parameter $\xi$. We are interested in the regime where $n$ scales linearly in $d$ and both are large such that the second term in the above equation is negligible.

*Remark 1. Why do we get the accuracy-on-the-line phenomenon?* The prediction of the trained (random) linear model on a random test data point $X$ involves the inner product, which performs a natural spatial averaging over the $d$ coordinates. Since the noise across the coordinates is independent and has bounded variance, the central limit theorem applies. The resulting error has a Gaussian tail. Hence, the probit mapping, $\Phi^{-1}(\mathrm{Acc}_{\theta_1\rho_1})$, is critical in translating accuracy into the relevant mean to standard deviation ratio: $\cos(\theta_1,\theta)\rho_1\rho\sqrt{n}/(\xi\sqrt{d})$. This is consequently important for getting the universal linear relation, because irrelevant parameters, $n$ and $\xi$, cancel out in the slope of $\Phi^{-1}(\mathrm{Acc}_{\theta_2\rho_2})/\Phi^{-1}(\mathrm{Acc}_{\theta_1\rho_1})$. Refer to the proof of Theorem 1 (Appendix F.1) for more details.

*Remark 2. Intersection at the random guess:* Previous work [43] that considered models with a wide range of accuracies has found that all lines intersect at a point corresponding to "random guess", which in this binary example is $(\Phi^{-1}(1/2), \Phi^{-1}(1/2)) = (0,0)$. Our theoretical analysis is consistent with this empirical observation: all universal lines intersect at $(0,0)$ up to a small additive error scaling as $O(1/\sqrt{n})$.

### 6.2 Input Mixing Yields an Intermediate Slope

Figures 3 and 4 show that when a model is trained on samples combined from two datasets, the resulting robustness trend lies in between the trends achieved by individual datasets. We show that this finding is universally true under our current setup in Theorem 2. Note that for the linear models we consider, input mixing (combining training sets) and output mixing (combining model outputs) are equivalent.

**Assumption 3.** *Consider two training datasets $\mathcal{D}_{n_1,\tilde{\theta}_1,\tilde{\rho}_1}$ and $\mathcal{D}_{n_2,\tilde{\theta}_2,\tilde{\rho}_2}$ of sizes $n_1$ and $n_2$ from distributions $P_{\tilde{\theta}_1,\tilde{\rho}_1}$ and $P_{\tilde{\theta}_2,\tilde{\rho}_2}$ as defined in Assumption 1. Separately training on individual datasets gives two models $\hat{\theta}_{n_1,\xi_1}(\mathcal{D}_{n_1,\tilde{\theta}_1,\tilde{\rho}_1})$ and $\hat{\theta}_{n_2,\xi_2}(\mathcal{D}_{n_2,\tilde{\theta}_2,\tilde{\rho}_2})$ as defined in Assumption 2. A model trained on a combined (and possibly subsampled) training dataset $\mathcal{D}_{n_1',\tilde{\theta}_1,\tilde{\rho}_1} \cup \mathcal{D}_{n_2',\tilde{\theta}_2,\tilde{\rho}_2}$ is represented by*

$$\hat{\theta}_{n_1'+n_2',\xi}(\mathcal{D}_{n_1',\tilde{\theta}_2,\tilde{\rho}_2} \cup \mathcal{D}_{n_2'\tilde{\theta}_2,\tilde{\rho}_2}) = \bar{\theta} + (\xi\|\theta\|/(\bar{\rho}\sqrt{n_1'+n_2'}))z \text{ where } \bar{\theta} = (n_1'\tilde{\theta}_1 + n_2'\tilde{\theta}_2)/(n_1' +$$

$n_2'$), $\bar{\rho} = \|\bar{\theta}\|/\sqrt{(n_1'\|\tilde{\theta}_1\|^2/\rho_1^2) + (n_2'\|\tilde{\theta}_2\|^2/\rho_2^2)}$, *and* $z \in \mathbb{R}^d$ *is a zero-mean random vector with independent entries each with variance one.*

Again, a canonical example of a trained model is $\hat{\theta}_{n_1'+n_2',\xi} = (1/(n_1'+n_2'))\big(\sum_{(x_i,y_i)\in\mathcal{D}_{n_1',\tilde{\theta}_1,\tilde{\rho}_1}} y_i x_i + \sum_{(x_i,y_i)\in\mathcal{D}_{n_2',\tilde{\theta}_2,\tilde{\rho}_2}} y_i x_i\big)$. We then have $\hat{\theta}_{n_1'+n_2',\xi} = \bar{\theta} + (\|\bar{\theta}\|/(\bar{\rho}\sqrt{n_1'+n_2'}))z$ with $\xi = 1$.

It follows from applying Theorem 1 to the model trained on the combined data (Assumption 3) that the resulting linear trend achieves

$$\mathrm{Slope}(\hat{\theta}_{n_1'+n_2',\xi}(\mathcal{D}_{n_1',\tilde{\theta}_1,\tilde{\rho}_1} \cup \mathcal{D}_{n_2',\tilde{\theta}_2,\tilde{\rho}_2})) \;:=\; \frac{\cos(\theta_2,\bar{\theta})\rho_2}{\cos(\theta_1,\bar{\theta})\rho_1}\,, \tag{3}$$

We show that the slope obtained from training on the combined dataset lies between those obtained from training on the two datasets separately. The proof could be found in Appendix F.2. Let $\mathrm{Slope}_1 := \mathrm{Slope}(\hat{\theta}_{n_1}(\mathcal{D}_{n_1,\tilde{\theta}_1,\tilde{\rho}_1}))$ and $\mathrm{Slope}_2$ be defined similarly.

**Theorem 2.** *Under Assumption 3,* $\mathrm{Slope}_1 \leq \mathrm{Slope}\big(\hat{\theta}_{n_1'+n_2'}(\mathcal{D}_{n_1',\tilde{\theta}_1,\tilde{\rho}_1} \cup \mathcal{D}_{n_2',\tilde{\theta}_2,\tilde{\rho}_2})\big) \leq \mathrm{Slope}_2$.

## 6.3   Filtering Data to Improve Robustness

The input mixing analysis in Section 6.2 suggests a filtering strategy to improve robustness.

**Assumption 4** (Gaussian distributions for filtering). *Consider a training dataset* $\mathcal{D}_{n,\theta_{\mathrm{train}},\rho}$ *of size* $n$ *draw i.i.d. from* $(X,Y) \sim P_{\theta_{\mathrm{train}},\rho}$ *where* $Y = \pm 1$ *uniformly at random and* $X|Y \sim \mathcal{N}(Y\theta_{\mathrm{train}}, (\|\theta_{\mathrm{train}}\|/\rho)^2\mathbf{I})$. *A model trained on an unfiltered* $\mathcal{D}_{n,\theta_{\mathrm{train}},\rho}$ *is denoted by* $\hat{\theta}_{\mathrm{unfiltered}} = (1/n)\sum_{(x_i,y_i)\in\mathcal{D}_{n,\theta_{\mathrm{train}},\rho}} x_i y_i$. *Note that* $\hat{\theta}_{\mathrm{unfiltered}} \sim \mathcal{N}(\theta_{\mathrm{train}}, (\|\theta_{\mathrm{train}}\|/(\rho\sqrt{n}))^2\mathbf{I})$.

*The model is evaluated on two test datasets: in-distribution (ID) and out-of-distribution (OOD), each with an isotropic Gaussian noise:* $(X,Y) \sim P_{\theta_{\mathrm{ID}},\rho_{\mathrm{ID}}}$, *where* $Y = \pm 1$ *uniformly at random and* $X|Y \sim \mathcal{N}(Y\theta_{\mathrm{ID}}, (\|\theta_{\mathrm{ID}}\|/\rho_{\mathrm{ID}})^2\mathbf{I})$, *and* $P_{\theta_{\mathrm{OOD}},\rho_{\mathrm{OOD}}}$ *is defined similarly. Let* $\mathrm{Slope}(\theta) := \frac{\cos(\theta_{\mathrm{OOD}},\theta)\rho_{\mathrm{OOD}}}{\cos(\theta_{\mathrm{ID}},\theta)\rho_{\mathrm{ID}}}$ *and assume that all models achieve better ID accuracy than OOD accuracy, i.e.,* $\mathrm{Slope}(\theta) < 1$. *A model is more robust if the slope is closer to one.*

Suppose we have access to a pre-trained model $\hat{\theta}_{\mathrm{pretrained}}$ that achieves better robustness than the model $\hat{\theta}_{\mathrm{unfiltered}}$ trained on the unfiltered data, i.e., $\mathrm{Slope}(\hat{\theta}_{\mathrm{unfiltered}}) < \mathrm{Slope}(\hat{\theta}_{\mathrm{pretrained}}) \leq 1$. We want to use the pre-trained model to filter the data $\mathcal{D}_{n,\theta_{\mathrm{train}},\rho}$ and achieve better robustness. We consider a generic family of filtering strategies that subsamples each data point $(x_i,y_i) \in \mathcal{D}_{n,\theta_{\mathrm{train}},\rho}$ with probability that is monotonically non-decreasing with its correlation to the pre-trained model, i.e., $\mathbb{P}((x_i,y_i) \text{ passes the filter}) \propto h(\langle x_i y_i, \hat{\theta}_{\mathrm{pretrained}}\rangle)$ for some monotonic scalar function $h : \mathbb{R} \to \mathbb{R}_+$. We analyze the model $\hat{\theta}_{\mathrm{filtered}} = (1/m)\sum_{(x_i,y_i)\in\text{filtered dataset}} x_i y_i$. The next theorem shows that any such filtering strategy improves robustness, and the full proof can be found in Appendix F.3.

**Theorem 3.** *Under Assumption 4, if* $\mathrm{Slope}(\hat{\theta}_{\mathrm{unfiltered}}) < \mathrm{Slope}(\hat{\theta}_{\mathrm{pretrained}}) \leq 1$ *then any model* $\hat{\theta}_{\mathrm{filtered}}$ *that is trained on the above family of filtered datasets achieves better robustness than the model trained on unfiltered data:* $\mathrm{Slope}(\hat{\theta}_{\mathrm{unfiltered}}) < \mathrm{Slope}(\mathbb{E}[\hat{\theta}_{\mathrm{filtered}}]) \leq \mathrm{Slope}(\hat{\theta}_{\mathrm{pretrained}})$, *where* $\mathrm{Slope}(\mathbb{E}[\hat{\theta}_{\mathrm{filtered}}])$ *denotes the slope of the linear trend of a model trained on the filtered data.*

*Remark 3. Relevance to empirical practice:* The data filtering approach currently employed for the LAION dataset [52] motivates our theoretical setup. The authors of LAION first retrieved a large number of image-text pairs from Common Crawl [1] and then computed the cosine similarity between the image and text embeddings using one of the CLIP models originally introduced by Radford et al. [48]. Next, the authors removed all pairs with similarity less than 0.3 from the dataset. While it was initially unclear if filtering the noisy Common Crawl source with an existing CLIP model would lead to a dataset with good generalization properties, experiments with medium-scale models on LAION so far demonstrate that the resulting models are comparable to the original CLIP models [2].

Our theorem 3 does not provide a full justification for the filtering process described above, but it offers theoretical evidence that filtering a noisy data source with an existing robust model can indeed improve the effective robustness of the resulting dataset. It remains to be shown empirically that

the slope of the linear trend exhibited by the CLIP model from [48] is larger than that of a model trained on the Common Crawl data pool without filtering. If this assumption holds, we can expect that training on data with better text-image alignment according to the original CLIP model, will improve robustness compared to training on unfiltered data from Common Crawl.

# 7 Conclusion

Motivated by CLIP's robustness to distribution shift, we introduced a testbed of six image-text datasets to study the influence of pre-training data on the robustness of multimodal models such as CLIP. We analyzed the interactions of these datasets through both experiments and theoretical analyses, finding that simply combining multiple datasets dilutes the robustness of the best-performing one. Moreover, we offer an explanation for the potential benefits of the data filtering process currently used for LAION. Our findings suggest that training on a carefully selected subset of data may provide more robustness than training on simply more data, e.g. obtained by combining multiple data sources.

To summarize, our results lead to the following recommendations for dataset design:

- On the evaluation front: building a pre-training dataset requires multiple robustness test sets, as different pre-training datasets exhibit vastly different behaviors across distribution shifts.
- On the dataset design front: each candidate data source for the overall pre-training dataset should be evaluated separately for its generalization and robustness properties. The final pre-training dataset should then contain more samples from the higher-quality pre-training sources. Creating a more "diverse" pre-training dataset by randomly sampling from all candidate sources does not result in a better dataset.
- Assuming access to models separately trained on each source of interest, output ensemble is a good predictor of the effective robustness obtained from input mixing, and hence can significantly reduce the time to search for the right mixing strategy.
- Filtering a noisy data source with an existing robust model can improve the generalization properties of the resulting dataset.

The broad societal implications of image-text models and web-crawled multimodal datasets have been extensively discussed by Radford et al. [48] and Birhane et al. [10] respectively. As these models continue to grow in size, full-scale experiments are becoming prohibitively expensive. This necessitates small, systematic experiments that allow reliable extrapolation to larger scales. Since effective robustness is independent of model size and quantity of training data, we hope that our testbed and results can serve as a first step towards building multimodal pre-training datasets in a principled manner. There remain interesting open questions about finding better ways to filter and combine samples from multiple data distributions, or how to build the most robust dataset given a fixed set of data sources. We look forward to addressing these in our future work.

**Limitations.** The pre-training datasets we experiment with range from 5M to 15M samples. While this lags behind current state-of-the-art settings that large image-text models are often trained on, we believe our findings also hold for models with higher accuracy for the following reasons:

- The original CLIP paper [48] offers evidence showing that CLIP's performance exhibits reliable linear trends across compute scales (Figure 9 of [48], which includes the ResNet50-encoder that our paper works with) and performance on natural distribution shifts (Figure 14 of [48]). In particular, the linear trends in this Figure 14 are analogous to the linear trends in our scatter plots.
- In the OpenCLIP GitHub repository [34] that seeks to match the performance of the original CLIP paper, the authors there also note that their models that were trained on 15M samples or fewer share the same effective robustness trend as OpenAI's CLIP models (trained on 400M samples). Refer to their Why are low-accuracy CLIP models interesting? remark for more details.
- Previous work [43] has experimented with training on randomly sampled subsets of the training dataset to produce models covering a wide range of accuracies, and found that these models lie on the same effective robustness trend. This includes models trained on only 1% of the original data, which matches the scale of our experiments (5M-15M samples) relative to the scale of the original CLIP paper (400M samples).

The focus of our work is to study the behavior of a diverse set of data sources, and performing full-scale experiments on 6 datasets (and their combinations) would be extremely expensive. This prompts us to leverage publicly available web-crawled datasets, and study the behavior of our trained models through the effective robustness framework, which has been shown to be scale-invariant.

## Acknowledgments and Disclosure of Funding

We would like to thank Tiffany Cai, Nicholas Carlini, Tatsunori Hashimoto, Jong Wook Kim, Hongseok Namkoong, Alec Radford, Rohan Taori, and Thomas Wolf for helpful discussions while working on this paper. We also thank Tiffany Cai and Jack Gallagher for their help with the dataset collection. HuggingFace has provided generous assistance in the form of compute resources that made many of our experiments possible. In particular, we would like to thank Leandro von Werra and Victor Sanh for their help with setting up the compute infrastructure and their constructive feedback on the dataset testbed. This work is supported in part by Open Philanthropy, the Allen Institute for AI, and NSF grants CNS-2002664, IIS-1929955, DMS-2134012, and CCF-2019844 as a part of NSF Institute for Foundations of Machine Learning (IFML).

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
