# Appendix

## A  Dataset Details

### A.1  Pre-training Datasets

| Dataset | Source | Total Size |
|---|---|---|
| YFCC | Flickr | 14,826,000 |
| LAION | Common Crawl | 15,504,742 |
| CC-12M | Unspecified web pages | 9,594,338 |
| RedCaps | Reddit | 11,882,403 |
| WIT | Wikipedia | 5,038,295 |
| ShutterStock | ShutterStock | 11,800,000 |

Table 1: Origin and total number of samples for each of the datasets we used in our experiments.

To get a better understanding of the diversity of different data sources, we analyze the distributions of caption lengths, image sizes and image aspect ratios for a set of 10,000 samples randomly selected from each source:

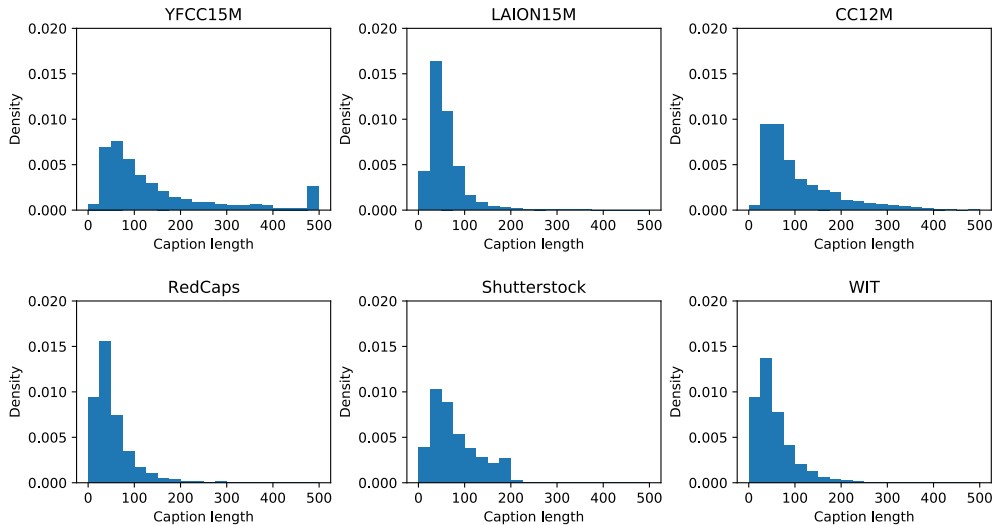

Figure 7: **Distributions of caption lengths for each data source.**

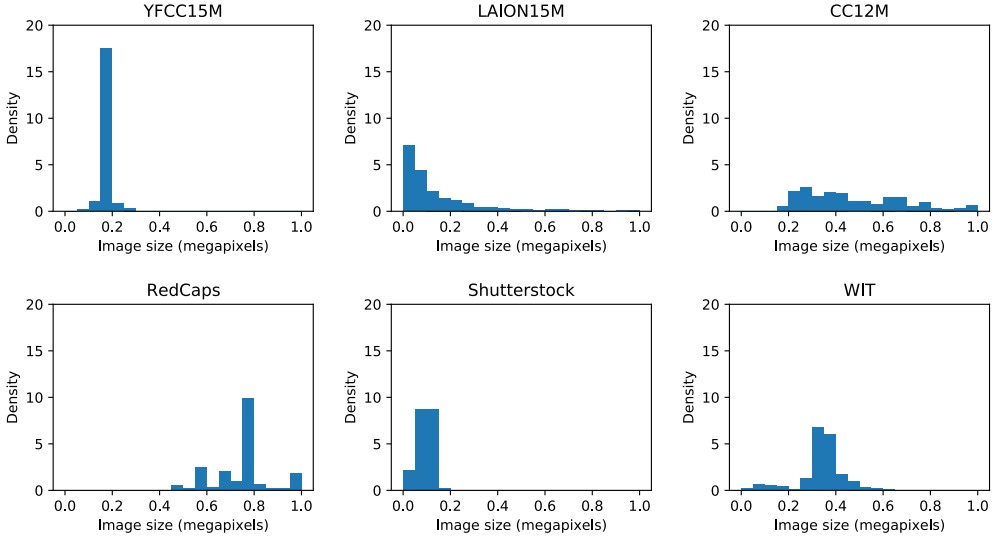

Figure 8: **Distributions of image sizes for each data source.**

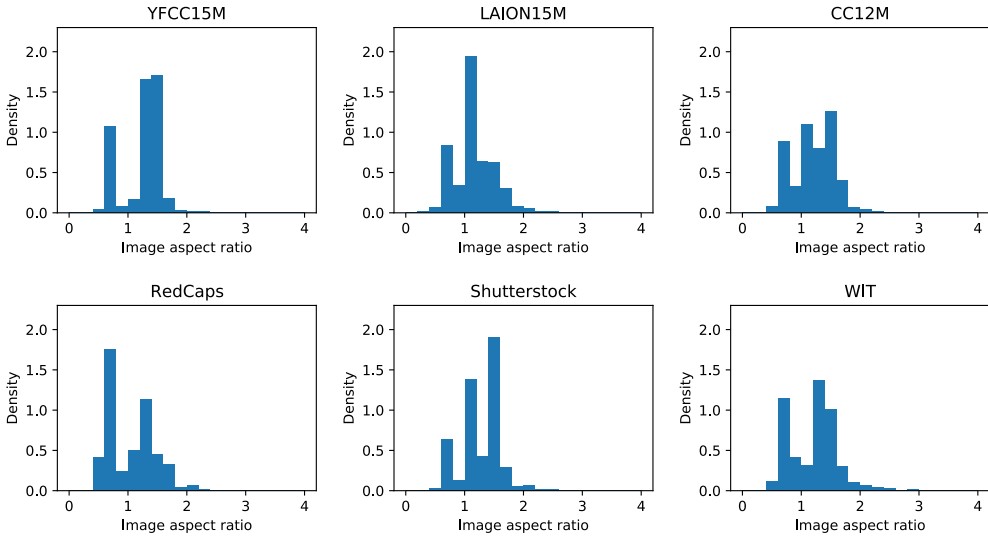

Figure 9: **Distributions of image aspect ratios for each data source.**

Below we also show some examples of image-caption pairs randomly selected from each data source:

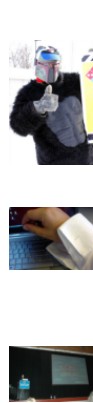 Cody This picture is #5 in my 100 strangers project. Find out more about the project and see pictures taken by other photographers at www.100Strangers.com.

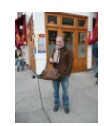 Dan with the Man-Purse Dan Budiac, metrosexual

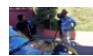 Cuffless To link or not to link, that is the question...

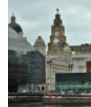 Dixie Union Chapel, 1836 Replaced a wooden structure that was built in 1816

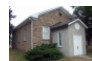 ATWS Slide presentation from Sandra Carvao WTO_0288 Slide presentation from Sandra Carvao Deputy Chief Market Intelligence and Promotion Department, World Tourism...

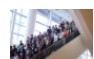 Web 2.0 Expo 2010 - San Francisco Please feel free to use this picture in your blog, website or presentation, in accordance with the stated Creative Commons and c...

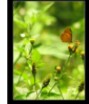 Look out, King Vidor Jesse behind the lens. Jodhpurs and bullhorn not included.

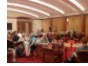 Liverpool, England, United Kingdom Albert Dock - Liverpool, England, United Kingdom

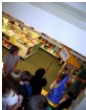 Mariposa Butterfly

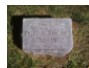 THATCamp Computational Archaeology August 2012, University of Virginia

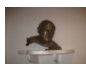 Harry Potter and the Half Blood Prince Release Party / September 16 They had to release this thing on my anniversary? Follow the flag to get your book. Auntie's ...

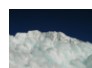 Nicholas Yeager Co. C, 1st Arkansas Infantry

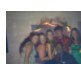 John Paul the Great St Patrick's Cathedral Charlotte NC May 20, 2007. We were there for the 33rd Annual Rosary Rally.

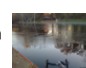 100 words for snow and ice A plethora of hues and textures

GNP: St. Mary Prom We had an employee prom in August. Great times in St. Mary :)

Ice at One Mile Low of 22 degrees on Tuesday, December 8th, 2009.

Figure 10: **Random training samples from YFCC.**

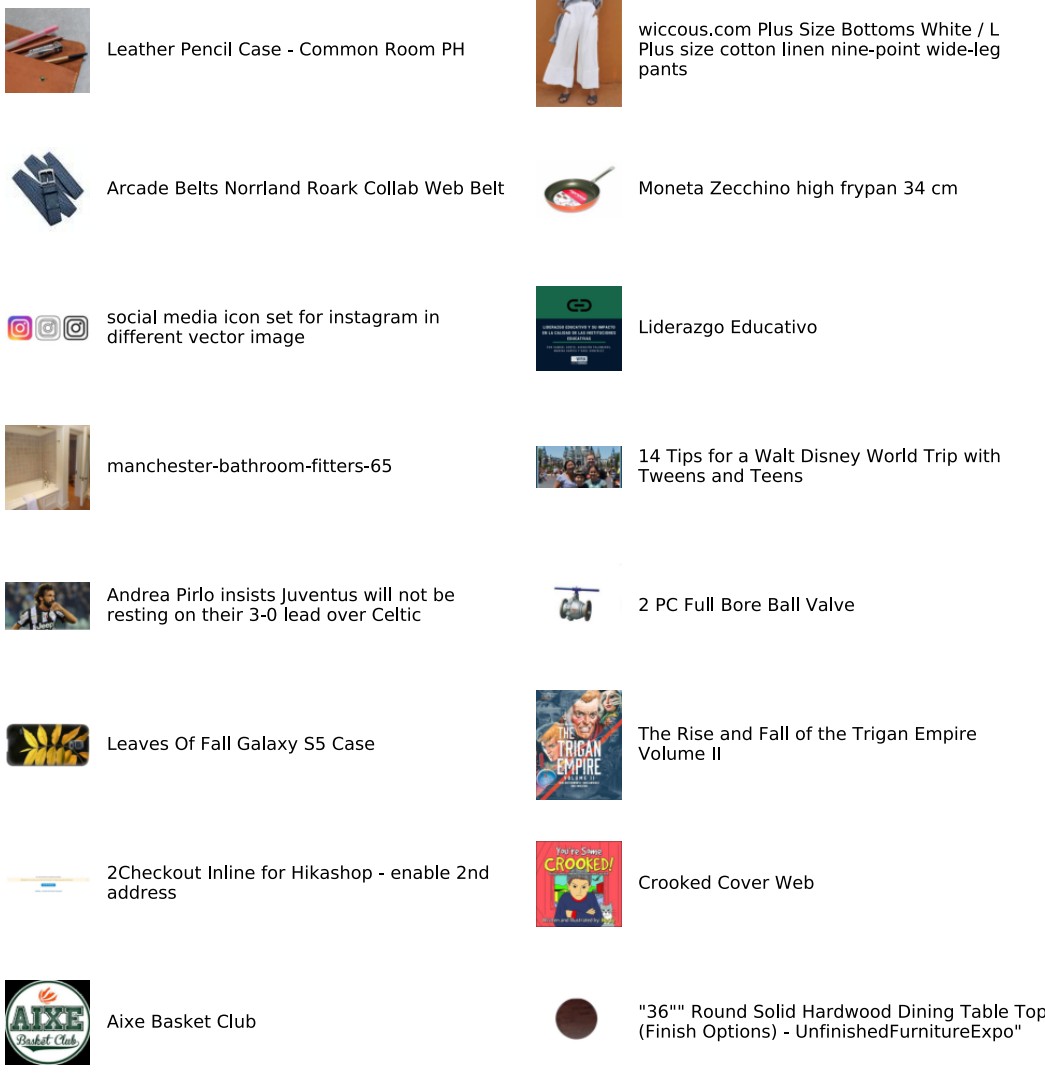

Leather Pencil Case - Common Room PH

wiccous.com Plus Size Bottoms White / L Plus size cotton linen nine-point wide-leg pants

Arcade Belts Norrland Roark Collab Web Belt

Moneta Zecchino high frypan 34 cm

social media icon set for instagram in different vector image

Liderazgo Educativo

manchester-bathroom-fitters-65

14 Tips for a Walt Disney World Trip with Tweens and Teens

Andrea Pirlo insists Juventus will not be resting on their 3-0 lead over Celtic

2 PC Full Bore Ball Valve

Leaves Of Fall Galaxy S5 Case

The Rise and Fall of the Trigan Empire Volume II

2Checkout Inline for Hikashop - enable 2nd address

Crooked Cover Web

Aixe Basket Club

"36"" Round Solid Hardwood Dining Table Top (Finish Options) - UnfinishedFurnitureExpo"

Figure 11: **Random training samples from LAION.**

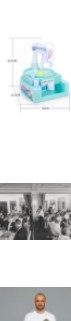 Wholesale toy candy machine for sale - Group buy The same Mini grabbing music clip candy machine small egg twisting machine grabbing childrens intellectual toys

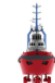 Tug Boat Model available on Turbo Squid, the world's leading provider of digital models for visualization, films, television, and games. Tug Boats, Motor Boats, B...

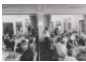 Real wedding RAC Epsom on the English Wedding Blog with Murray Clarke Photography (62)

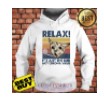 Cat smoking relax it's just the bud not corona virus vintage s Hoodie

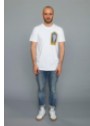 The North Hill <PERSON> is a made in France t-shirt

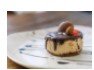 Cake on a white plate, presentation.

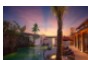 The swimming pool at or near Maca Villas and Spa

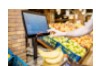 Weighing bananas in the supermarket. Man weighing bananas on the scales in the supermarket, close-up view with no face stock images

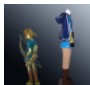 The Legend of Zelda Breath of The Wild Link Costume

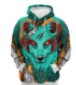 The Green Elite Wolf Hoodie

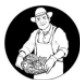 Farmer with basket vegetables, isolated in a round frame, contour drawing, icon, logo, coloring, black and white vector. Illustration, outline cartoon drawing sto...

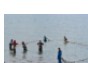 Students in waders stand in ocean water with a large floating net.

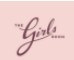 The Girls Rooms logo design type logo layout branding design

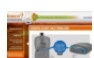 Each page explains an aspect of the product, with easy navigation and animation to help drive the point across.

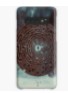 The Maze Samsung Galaxy Snap Case

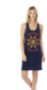 Every Day Is Another Chance Sun Women's Cotton Modal Jersey Tank Dress

Figure 12: **Random training samples from Conceptual Captions (CC-12M).**

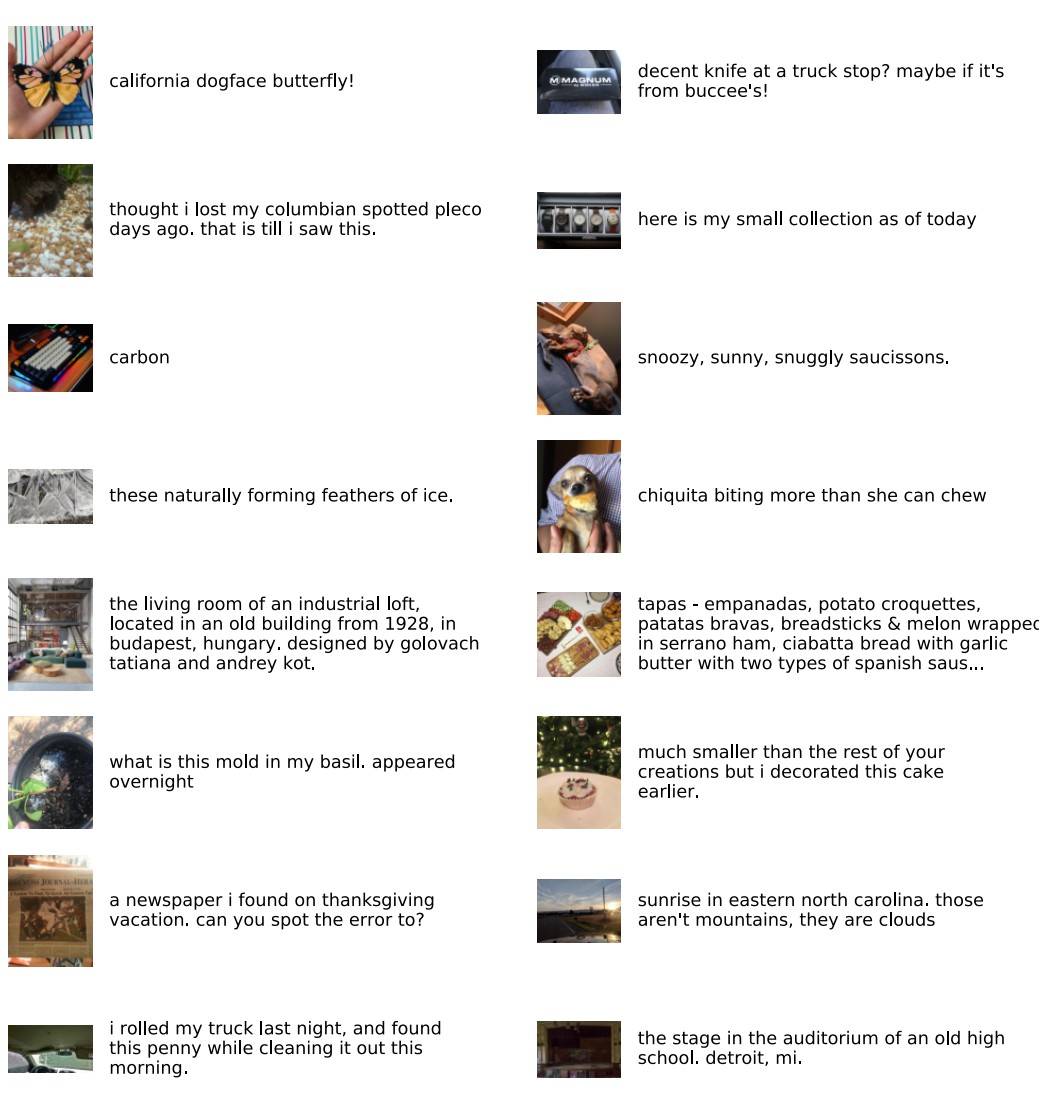

Figure 13: **Random training samples from RedCaps.**

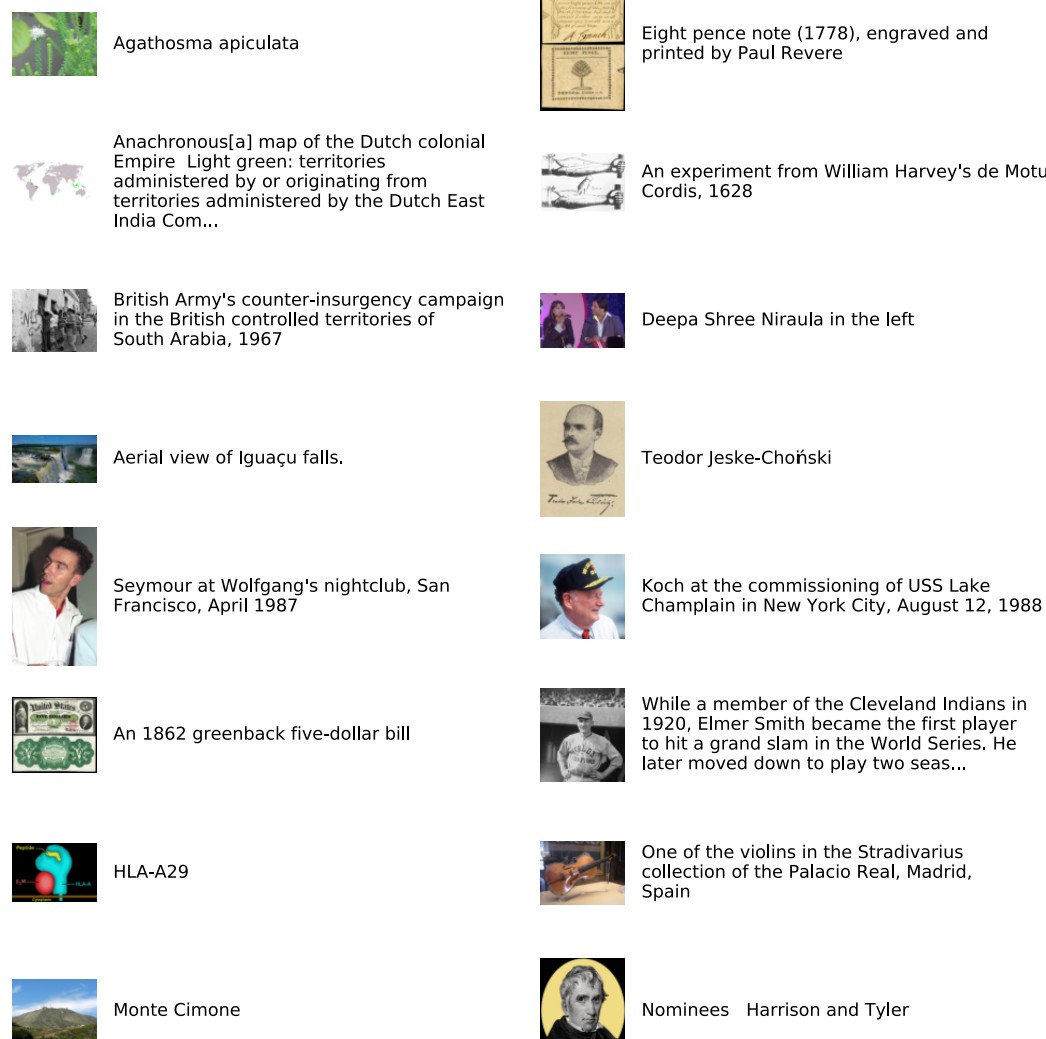

Figure 14: **Random training samples from WIT.**

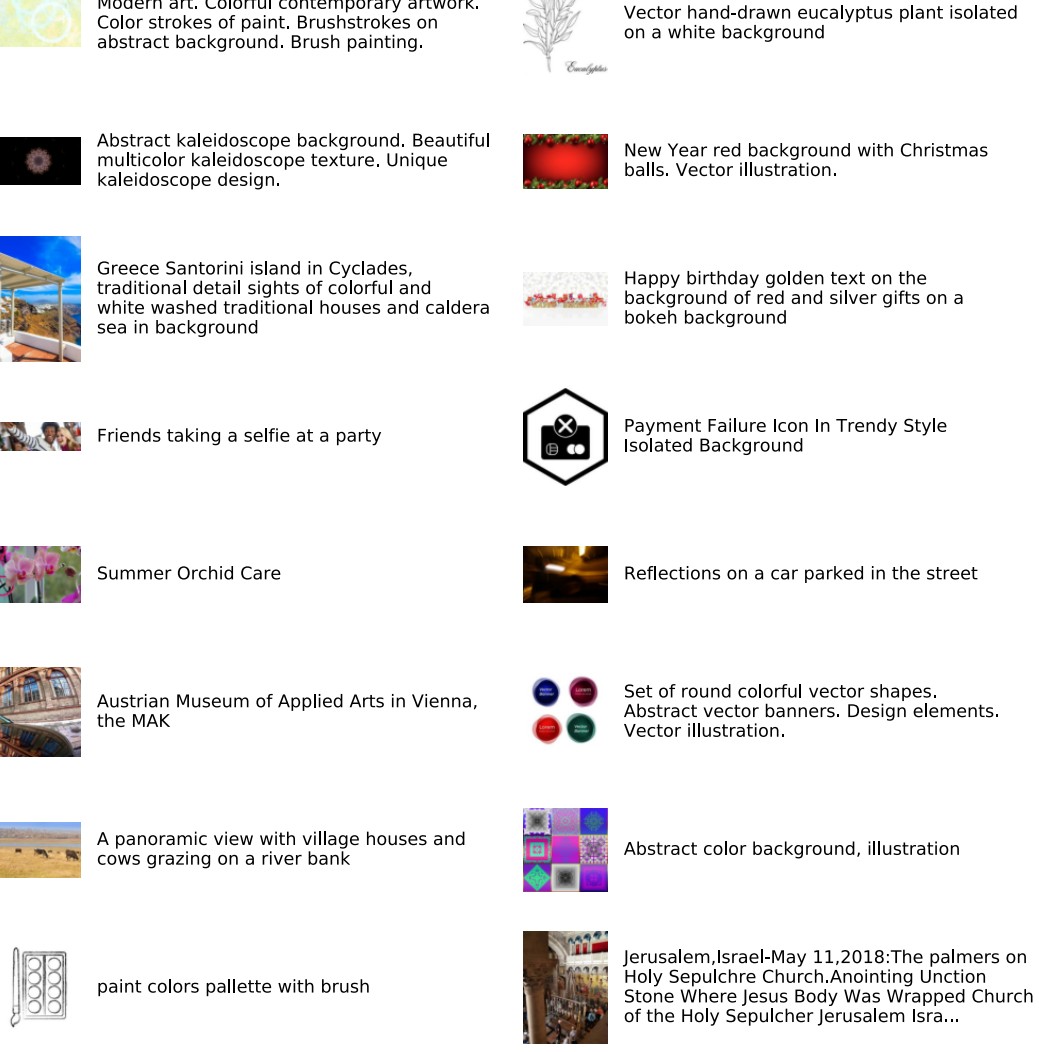

Modern art. Colorful contemporary artwork. Color strokes of paint. Brushstrokes on abstract background. Brush painting.

Vector hand-drawn eucalyptus plant isolated on a white background

Abstract kaleidoscope background. Beautiful multicolor kaleidoscope texture. Unique kaleidoscope design.

New Year red background with Christmas balls. Vector illustration.

Greece Santorini island in Cyclades, traditional detail sights of colorful and white washed traditional houses and caldera sea in background

Happy birthday golden text on the background of red and silver gifts on a bokeh background

Friends taking a selfie at a party

Payment Failure Icon In Trendy Style Isolated Background

Summer Orchid Care

Reflections on a car parked in the street

Austrian Museum of Applied Arts in Vienna, the MAK

Set of round colorful vector shapes. Abstract vector banners. Design elements. Vector illustration.

A panoramic view with village houses and cows grazing on a river bank

Abstract color background, illustration

paint colors pallette with brush

Jerusalem,Israel-May 11,2018:The palmers on Holy Sepulchre Church.Anointing Unction Stone Where Jesus Body Was Wrapped Church of the Holy Sepulcher Jerusalem Isra...

Figure 15: **Random training samples from ShutterStock.**

### A.2 Test Distributions

Figure 16 illustrates the four distribution shifts that we use for evaluating the quality of CLIP features after pre-training on different data sources.

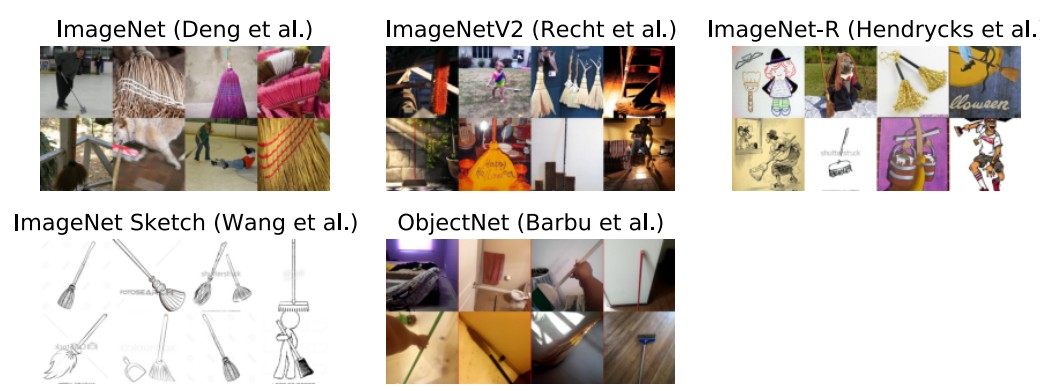

Figure 16: **Distribution shifts at test time.** We visualize samples of the class "broom" from the reference distribution ImageNet [18], and the four distribution shifts derived from ImageNet: ImageNet-V2 [50], ImageNet-R [32], ImageNet-Sketch [59] and ObjectNet [6].

## B Training Details

Our implementation closely follows the training code from OpenCLIP GitHub repository [34]. When training CLIP from scratch on each of the pre-training datasets, unless otherwise mentioned, we use AdamW optimizer [42] with default PyTorch parameters $\beta_1 = 0.9, \beta_2 = 0.999, \epsilon = 10^{-8}$, (per GPU) batch size 128 and weight decay of 0.1. For learning rate, we start with a learning rate of $10^{-3}$ and apply a cosine-annealing learning rate schedule [41] with 5,000 steps. We use the same data augmentations as in [48]. Models then undergo distributed training on 8 A40 or A100 GPUs for 16 epochs.

# C   Behavior of Individual Data Sources

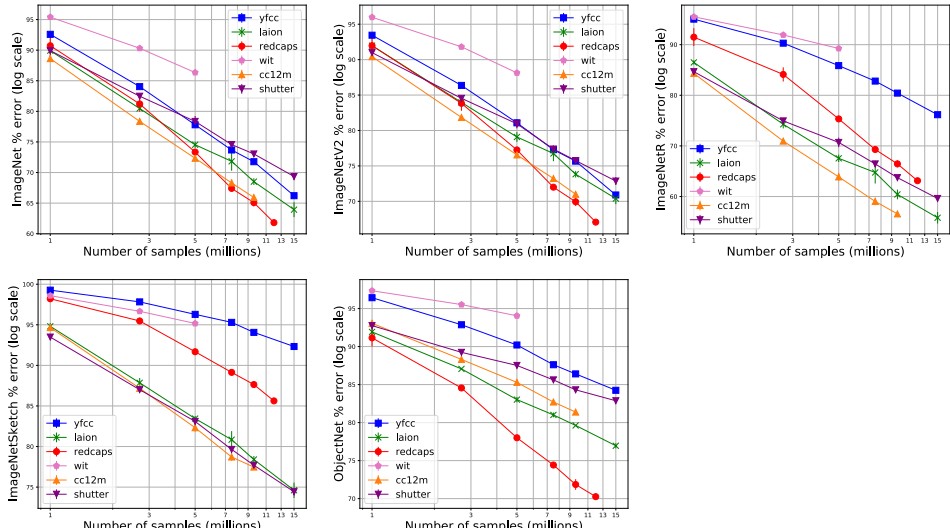

Figure 17: **Data efficiency of the six pre-training data sources on different test sets**. For each source, we randomly sample various subsets of data with sizes ranging from 1M to a maximum of 15M samples, and measure the zero-shot classification error of a CLIP model trained on the subset, on ImageNet and the four shifted test sets (i.e., ImageNet-V2, ImageNet-R, ImageNet-Sketch, ObjectNet). Plotted error values are log-transformed and averaged over 3 random seeds. We find that the data efficiency (i.e., how fast the error would decrease with more samples) of the six data sources varies significantly based on the evaluation setting.

# D Input Mixing

## D.1 More Experiments with CLIP Pre-training Data Sources

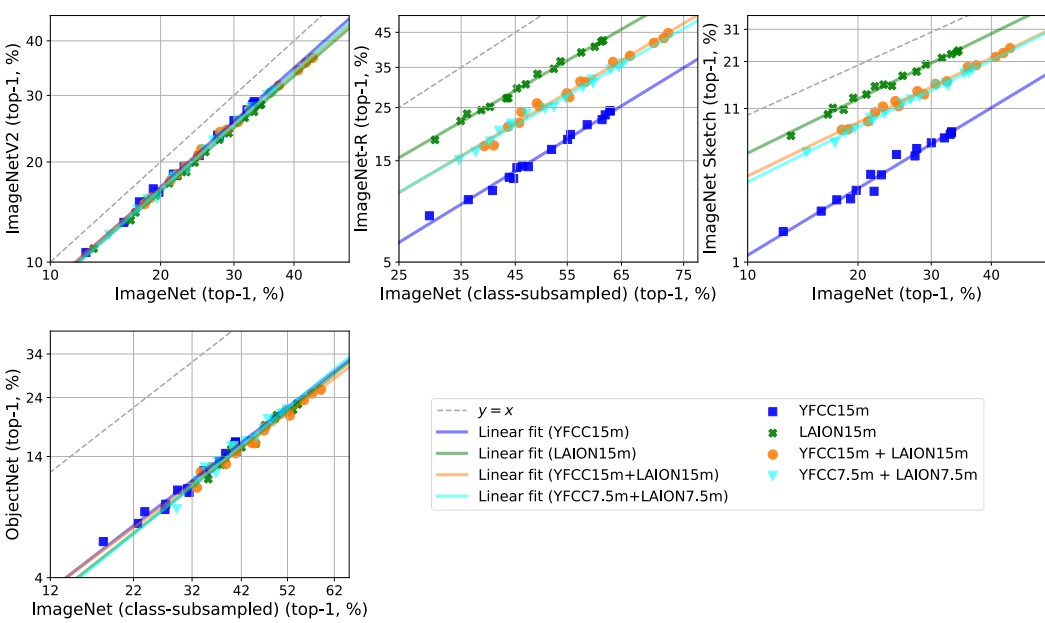

Figure 18: **Full plot for Figure 3 with all distribution shifts.** Combining YFCC and LAION training data in equal ratios results in a CLIP model with intermediate robustness.

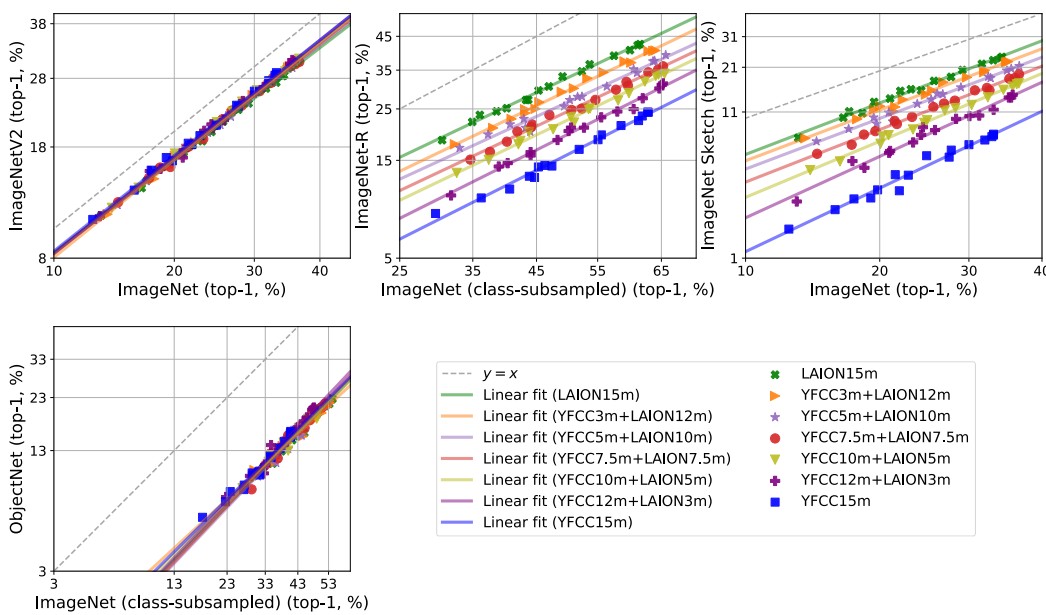

Figure 19: **Full plot for Figure 4 with all distribution shifts.** Varying the sample contributions of YFCC and LAION to the input data mixture produces a smooth interpolation of the linear trend between the trends of training on YFCC and LAION separately.

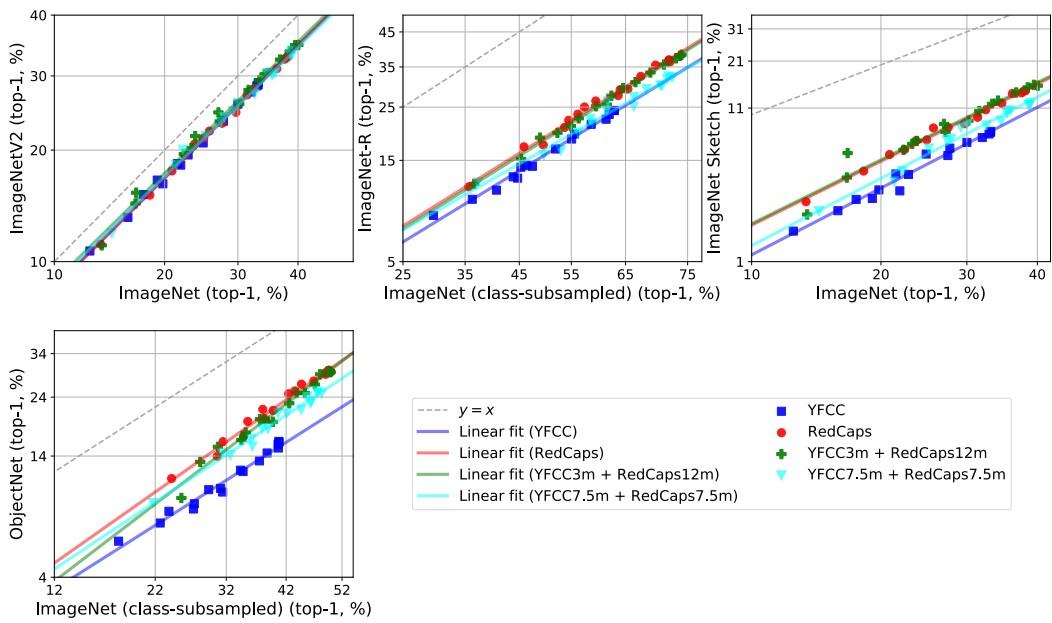

Figure 20: **Input mixing results for YFCC and RedCaps data sources.** Similar to previous observations (Figure 4), combining YFCC and RedCaps data in the pre-training dataset with different ratios yields different linear trends that all lie between that of training on YFCC and that of training on RedCaps alone.

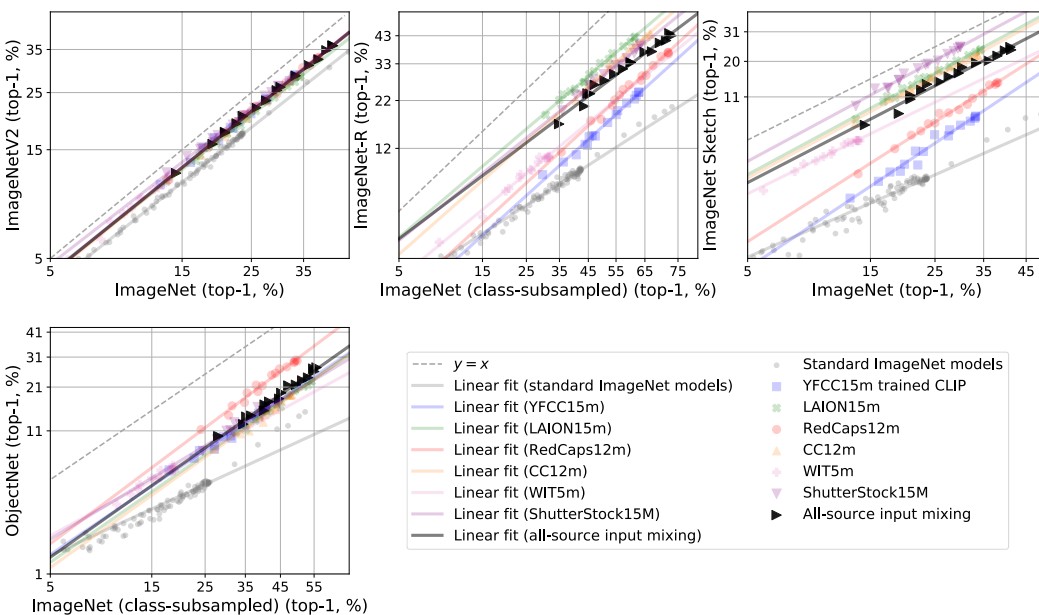

Figure 21: **Input mixing results for all six data sources.** We combine data from all six sources in the testbed with equal ratios (i.e., taking 2.7M samples from each), and find that the resulting robustness of CLIP trained on this data mixture (black line), is less than that of training only on the best-performing data source for each distribution shift setting.

## D.2 Experiments on CIFAR-10 & CINIC-10

We also investigate the phenomenon that mixing data sources resulting in diluted robustness (Section 5.1) in smaller-scale, uni-modal classification settings. Here, we experiment with mixing CIFAR-10 [37] and CINIC-10 [17] sources, each having 50K samples in total. CINIC-10 is itself a mixture of CIFAR-10 images and images selected and downsampled from the ImageNet database (for the same 10 classes). We use three architectures—ResNet-18, ResNet-34 and ResNet-50 [28]—and vary the number of epochs of training to obtain models of different accuracies. Models are evaluated on both CIFAR-10 and CINIC-10 standard test sets, and their performances are plotted along the axes of a scatter plot. Similar to previous input mixing results for CLIP, we observe in Figure 22 that ResNets trained on a 50K-sample dataset made up of *both* CIFAR-10 and CINIC-10 data, produce a linear trend that lies in between the trends of training models separately on just 50K CIFAR-10 images and just 50K CINIC-10 images.

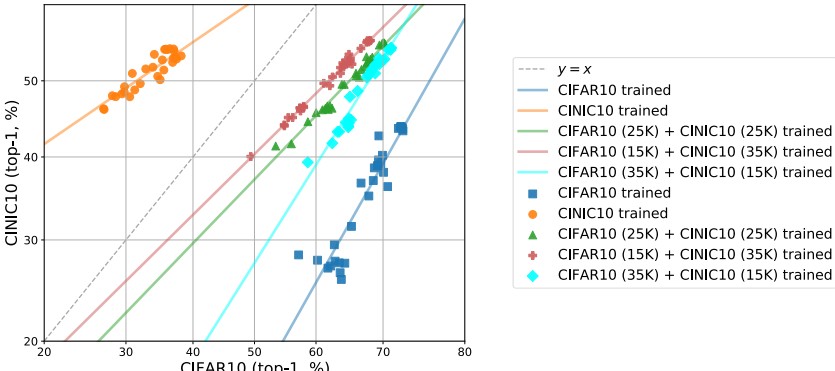

Figure 22: **Mixing inputs from CIFAR-10 and CINIC-10 distributions also produces models with intermediate robustness.** Similar to our findings from the multimodal setting with CLIP pre-training, we also observe that for standard image classification tasks like CIFAR-10 and CINIC-10, combining data samples from these two distributions with varying ratios ends up diluting the robustness of the original sources. The training set size is fixed at 50K samples for all linear trends displayed in this plot.

# E    Output Mixing

## E.1    More Experiments with CLIP Pre-training Data Sources

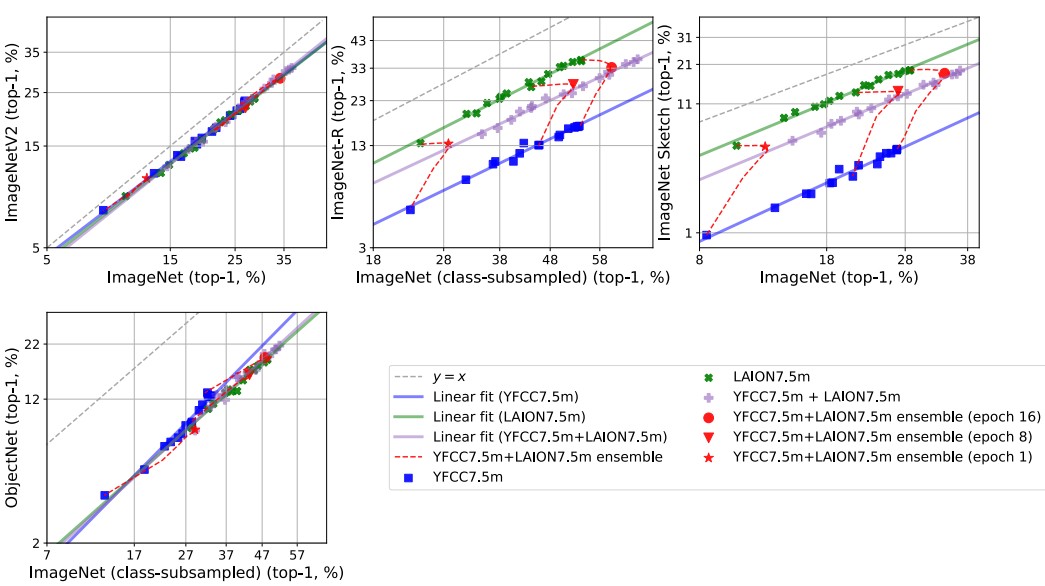

Figure 23: **Full plot for Figure 5 with all distribution shifts.** Ensemble outputs of two CLIP models trained on YFCC and LAION separately share the same linear trend as a *single* model trained on the combined data mixture (with equal sample contribution from each source).

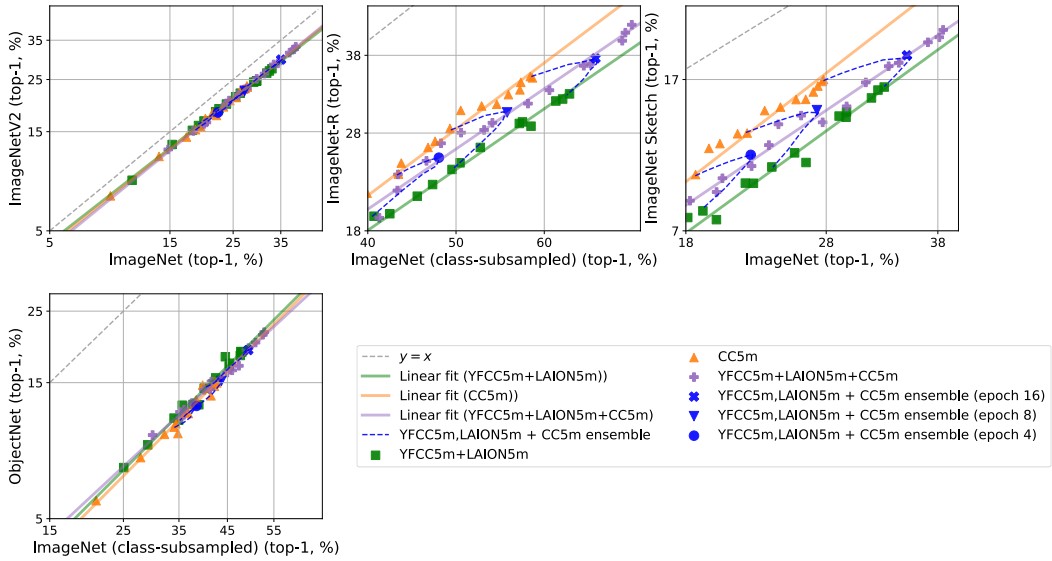

Figure 24: **Full plot for Figure 6 with all distribution shifts.** Given an existing pre-training dataset that could be a mixture (e.g., YFCC-5M + LAION-5M, green line) and a new data source (e.g., CC-5M, orange line), we could use the ensemble outputs (blue markers) of two CLIP models that have been trained separately on these two data distributions, to estimate the linear trend for models that would be trained on *all* the data (purple line).

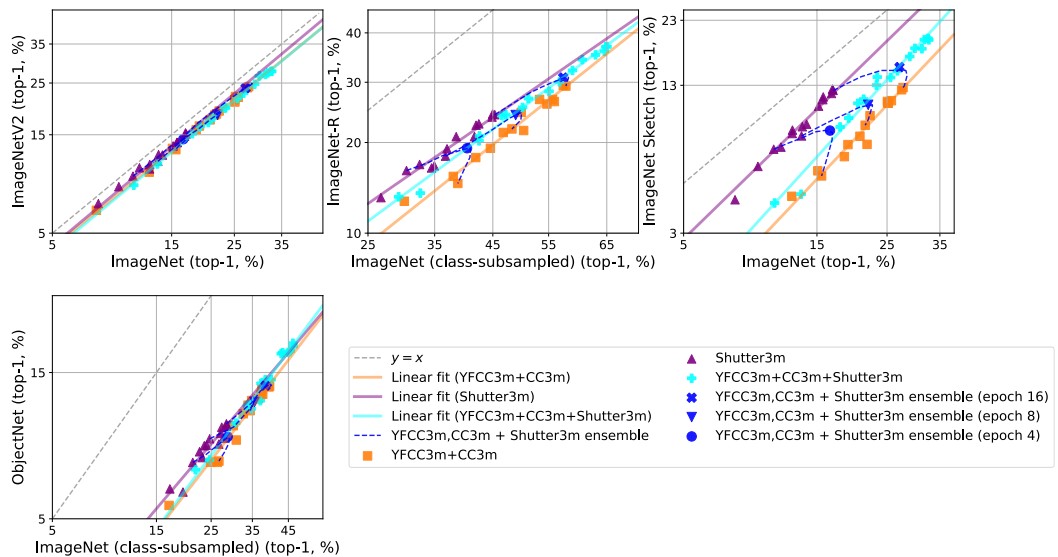

Figure 25: **Output mixing results for two CLIP models trained on YFCC-3M + CC-3M mixture and ShutterStock-3M respectively**. We repeat the experiment in Figure 24 for a different set of data sources (YFCC, ShutterStock, Conceptual Captions), taking 3M samples from each. The same output mixing phenomenon applies: the ensemble outputs of CLIPs trained on different data sources and dataset sizes (purple and orange lines), taken from the same epoch, lie on the linear trend of training a single model on the combined dataset made up of these three sources (cyan line). The two models' logit predictions are ensembled with equal weights (blue markers).

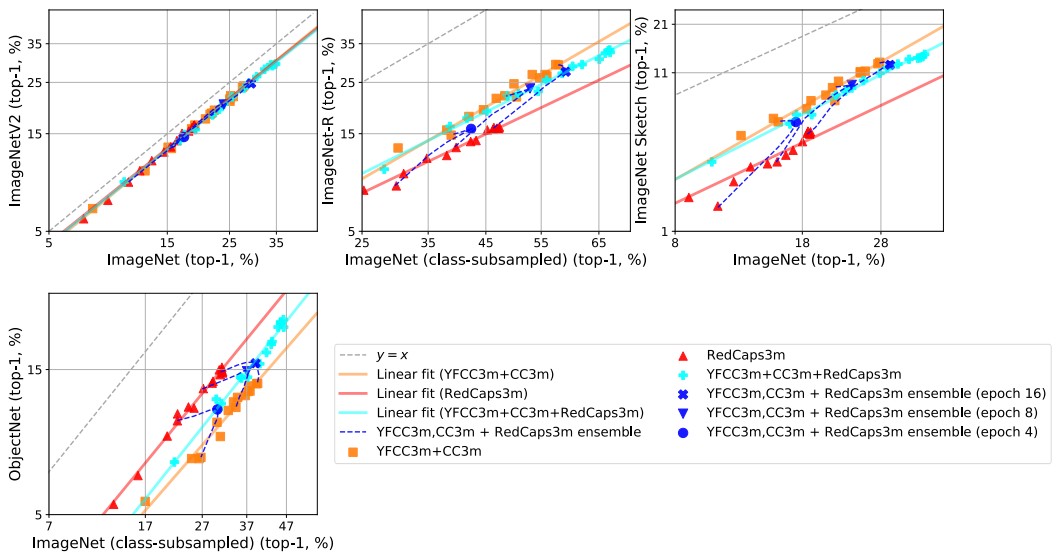

Figure 26: **Output mixing results for two CLIP models trained on YFCC-3M + CC-3M mixture and RedCaps-3M respectively**. Ensemble outputs of CLIPs trained on different data sources and dataset sizes (red and orange lines), taken from the same stage of training (i.e., epoch), lie on the linear trend of training a single model on the combined dataset made up of these three sources (cyan line), when the two models' logit predictions are ensembled with equal weights (blue markers).

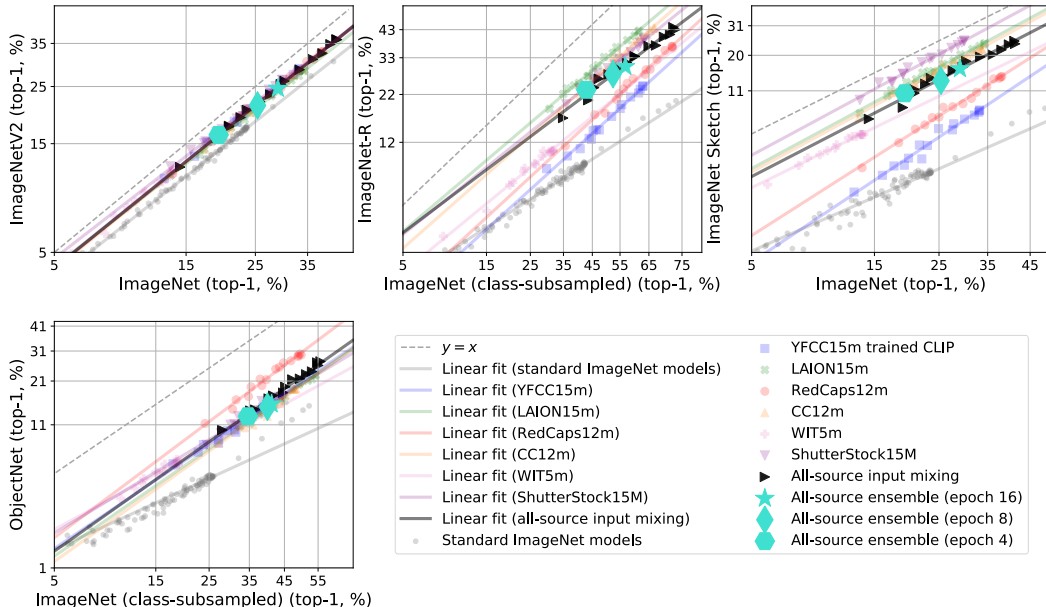

Figure 27: **Ensemble outputs of CLIPs trained separately on each of the data sources of interest share the same linear trend as a single CLIP model trained on the 6-source data mixture.** Following the input mixing setup in Figure 21, when we ensemble the logit predictions of six CLIP models, each trained on 2.7M samples randomly selected from a *single* data source, with equal weights, we find that the ensemble outputs are also predictive of the linear trend of training CLIP models on a *single* data mixture made up of 2.7M samples from each source.

## E.2 Experiments on CIFAR-10 & CINIC-10

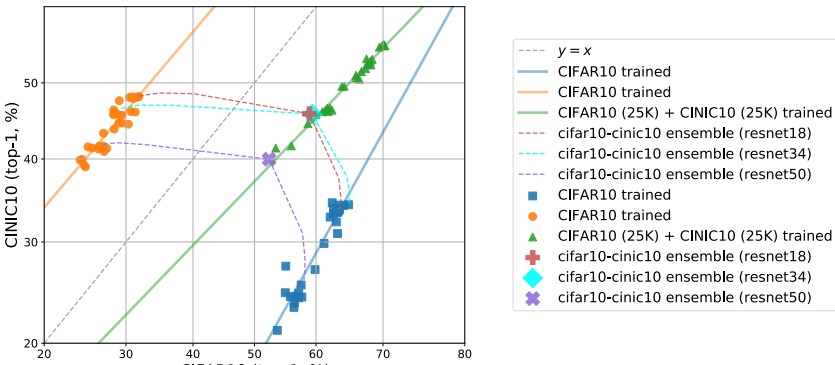

Figure 28: **Ensembling outputs of two models trained separately on CIFAR10 and CINIC10 lie on the same linear trend as training from scratch on the combined data mixture (where each source contributed equally).** We combine the logit predictions of CINIC10-trained and CIFAR10-trained models that have the same architecture (e.g., ResNet-18, ResNet-34 and ResNet-50 in this case) with varying ensemble weights between 0 and 1 (dashed lines). Similar to our findings from the multimodal setting with CLIP, we also observe that when the predictions are combined with equal weights (markers on the dashed lines), the resulting test accuracies on the two corresponding test sets lie on the linear trend produced by training ResNets on a CIFAR10 + CINIC10 data mixture with equal number of samples from each source.

# F Proofs of the Analyses

We provide proofs of main theoretical claims in Section 6.

## F.1 Proof of Theorem 1

**Assumption 5.** *Under the hypotheses of Theorem 1, suppose there exists a positive constant $c$ such that the third moments are bounded by $\mathbb{E}_{(X,Y)\sim P_{\theta_1,\rho_1}}[(X_iY - \theta_{1,i})^3] \leq c\mathbb{E}_{(X,Y)\sim P_{\theta_1,\rho_1}}[(X_iY - \theta_{1,i})^2]^{3/2}$, $\mathbb{E}_{(X,Y)\sim P_{\theta_2,\rho_2}}[(X_iY-\theta_{1,i})^3] \leq c\mathbb{E}_{(X,Y)\sim P_{\theta_2,\rho_2}}[(X_iY-\theta_{1,i})^2]^{3/2}$, and $\mathbb{E}[(\hat{\theta}_{n,i}-\theta_i)^3] \leq c\mathbb{E}[(\hat{\theta}_{n,i} - \theta_i)^2]^{3/2}$ for all $i \in [d]$.*

Under this assumption, we show that

$$\Phi^{-1}(\mathrm{Acc}_{\theta_1,\rho_1}) = \cos(\theta_1, \theta)\rho_1\frac{\rho}{\xi}\sqrt{\frac{n}{d}} + O\Big(\frac{\exp(\frac{\rho_1^2\rho^2 n}{2\xi^2 d})}{\sqrt{n}}\Big), \text{ and} \tag{4}$$

$$\Phi^{-1}(\mathrm{Acc}_{\theta_2,\rho_2}) = \cos(\theta_2, \theta)\rho_2\frac{\rho}{\xi}\sqrt{\frac{n}{d}} + O\Big(\frac{\exp(\frac{\rho_2^2\rho^2 n}{2\xi^2 d})}{\sqrt{n}}\Big), \tag{5}$$

as it will make the first and second claims straightforward. For $(X_1, Y_1) \sim P_{\theta_1,\rho_1}$, the first error event is $\{\mathrm{sign}(\langle X_1, \hat{\theta}_{n,\xi}\rangle) \neq Y_1\} = \{\langle X_1, \hat{\theta}_{n,\xi}\rangle Y_1 \leq 0\} = \{\langle\theta+(\xi\|\theta\|/\rho\sqrt{n})z, \theta_1+(\|\theta_1\|/\rho_1)z_1\rangle \leq 0\}$, where we used the fact that $\hat{\theta}_{n,\xi} = \theta + (\xi\|\theta\|/\rho\sqrt{n})z$ and $XY \stackrel{d}{=} \theta_1 + (\|\theta_1\|/\rho_1)z_1$. Since the third moments are bounded, applying Berry-Esseen theorem, we get that the probability of error is bounded by $\Phi(-(\langle\theta,\theta_1\rangle\rho_1\rho\sqrt{n}/(\xi\|\theta_1\|\|\theta\|\sqrt{d}))) + O(1/\sqrt{d})$. This gives $\mathrm{Acc}_{\theta_1,\rho_1} = \Phi(\langle\theta,\theta_1\rangle\rho_1\rho\sqrt{n}/(\xi\|\theta_1\|\|\theta\|\sqrt{d})) + O(1/\sqrt{d})$, and consequently

$$\Phi^{-1}(\mathrm{Acc}_{\theta_1,\rho_1}) = \cos(\theta_1, \theta)\,\rho_1\,\frac{\rho}{\xi}\sqrt{\frac{n}{d}} + O\Big(\frac{e^{\frac{\cos(\theta_1,\theta)^2\rho_1^2\rho^2 n}{2\xi^2 d}}}{\sqrt{n}}\Big). \tag{6}$$

This proves the desired claim.

## F.2 Proof of Theorem 2

Recall that $\mathrm{Slope}(\hat{\theta}_n(\mathcal{D}_{n,\theta,\rho})) = c\langle\theta_2, \theta\rangle/\langle\theta_1, \theta\rangle$ for a positive constant $c > 0$ that does *not* depend on the training data. Although the slope only depends on the training data and algorithm through $\theta$, we write all the parameters including the sample size $n$ and the training SNR $\rho$ to make it explicit that the results hold for all variations of the sample size and the training algorithm within the class that we assume. It is sufficient to show that this is a monotonic function over $\theta$ when we linearly traverse from $\tilde{\theta}_1$ to $\tilde{\theta}_2$, i.e. $\theta(\alpha) = \alpha\tilde{\theta}_1 + (1 - \alpha)\tilde{\theta}_2$ for $\alpha \in [0,1]$. Note that $f(\alpha) = c\langle\theta_2, \theta(\alpha)\rangle/\langle\theta_1, \theta(\alpha)\rangle = c_1 + c_2/\langle\theta_1, \theta(\alpha)\rangle$ for some $c_1$ and $c_2$ that do not depend on $\alpha$. The monotonicity follows from the fact that the derivative is

$$\frac{\partial f(\alpha)}{\partial\alpha} = -c_2\frac{\langle\theta_1, \tilde{\theta}_1 - \tilde{\theta}_2\rangle}{\langle\theta_1, \theta(\alpha)\rangle^2},$$

whose sign does not change for any $\alpha$.

## F.3 Proof of Theorem 3

The train data distribution satisfies $x_iy_i \sim \mathcal{N}(\theta_{\mathrm{train}}, (\|\theta_{\mathrm{train}}\|/\rho)^2\mathbf{I})$. Note that filtering does not change the distribution in $d - 1$ dimensional subspace orthogonal to $\hat{\theta}_{\mathrm{pretrained}}$, due to rotation invariance of a spherical Gaussian distribution. This implies that $\mathbb{E}[\mathcal{P}_\perp(\hat{\theta}_{\mathrm{filtered}})] = \mathcal{P}_\perp(\theta_{\mathrm{train}})$, where $\mathcal{P}_\perp$ denotes the projection operator to the $d-1$ dimensional subspace orthogonal to $\hat{\theta}_{\mathrm{pretrained}}$. On the other hand, on the direction of $\hat{\theta}_{\mathrm{pretrained}}$, the filtering increases the correlation in expectation: $|\mathbb{E}[\mathcal{P}_\|(\hat{\theta}_{\mathrm{filtered}})] - \mathcal{P}_\|(\hat{\theta}_{\mathrm{pretrained}})| \leq |\mathcal{P}_\|(\theta_{\mathrm{train}}) - \mathcal{P}_\|(\hat{\theta}_{\mathrm{pretrained}})|$, where $\mathcal{P}_\|$ denotes the projection operator to the one dimensional subspace spanned by $\hat{\theta}_{\mathrm{pretrained}}$. This implies that $\mathbb{E}[\hat{\theta}_{\mathrm{filtered}}] = \theta_{\mathrm{train}} + c\hat{\theta}_{\mathrm{pretrained}}$ for some positive $c$. It follows that $\mathbb{E}[\hat{\theta}_{\mathrm{filtered}}]$ is a convex interpolation between

two vectors, each in the direction of $\theta_{\mathrm{trian}}$ and $\hat{\theta}_{\mathrm{pretrained}}$, respectively. We can apply Theorem  which gives that

$$\mathrm{Slope}(\hat{\theta}_{\mathrm{unfiltered}}) < \mathrm{Slope}(\mathbb{E}[\hat{\theta}_{\mathrm{filtered}}]) \leq \mathrm{Slope}(\hat{\theta}_{\mathrm{pretrained}}) \ ,$$

when $\mathrm{Slope}(\hat{\theta}_{\mathrm{unfiltered}}) < \mathrm{Slope}(\hat{\theta}_{\mathrm{pretrained}})$.