# OpenReview forum: "Quality Not Quantity: On the Interaction between Dataset Design and Robustness of CLIP"
_NeurIPS.cc/2022/Conference — NeurIPS 2022 Accept_

### Official Review · Reviewer_Y6Mj · 2022-07-10

**Rating:** 8
**Confidence:** 5
**Soundness:** 4 excellent
**Presentation:** 4 excellent
**Contribution:** 4 excellent

**Summary:**

This paper studies the role of training data on the robustness of large image-text models like CLIP and ALIGN.
The paper is structured into two main parts — empirical analysis, and theoretical discussion.
The empirical analysis involves training CLIP models using six different datasets containing web-curated image-text pairs, and report their robustness on existing testbeds covering ImageNet classes.
The second half of the paper presents a theoretical discussion considering a binary classification setup, to explain empirical results.
Taken together, the paper argues that naive scaling of training data is not sufficient to improve robustness -- careful consideration during data curation is critical to endow the trained models with the desired robustness.

**Questions:**

The experimental setup makes heavy use of the recent LAION dataset.
A recent audit of the LAION dataset found several ethical issues with the dataset, e.g. verifiably pornographic imagery (especially of marginalized communities).
See "[Multimodal datasets: misogyny, pornography, and malignant stereotypes](https://arxiv.org/abs/2110.01963)" for a more comprehensive discussion.

The authors have applied basic filtering to avoid NSFW content, which is a great step -- I suggest citing this dataset audit paper and making a direct note in the paper to ensure that future works also apply similar filtering to minimize over-amplification of problematic imagery.
I also suggest releasing all trained models to aid follow-up works and increase the potential impact on the community.

**Limitations:**

**Discussion about limitations and societal impact is adequate (Section 7).**
I concur with the authors' assessment – this paper focuses on analysis and understanding of CLIP (and similar models), it is fair to say that its societal impacts are not very different/more than CLIP itself.

**Strengths And Weaknesses:**

I think this paper, in its current form, already matches the quality of a typical publication at NeurIPS conference. I strongly recommend acceptance. Below I justify my rating by discussing the main strengths of the paper along each of the suggested axes:

**Originality:**
Large pre-trained models like CLIP and ALIGN are becoming increasingly popular in the vision community.
While most works make progress by scaling up data and models (e.g. ALIGN, BASIC) or by adding dense supervision (e.g. SLIP, DeCLIP, FILIP), this paper focuses on the impact of training data source/distribution on certain downstream tasks.
Experiments study CLIP's robustness conditioned on its training data, and theoretical discussion attempts to explain results obtained using noisy, internet-scale data.
To the best of my knowledge, this exploration using large pre-trained models at this level of detail is novel.

Side note: Findings of this paper match with a recent paper — [Data Determines Distributional Robustness in Contrastive Language Image Pre-training (CLIP)](https://arxiv.org/abs/2205.01397). However, the experiments in this paper are different and the theoretical discussion is new. Moreover, this paper appeared on arxiv very close to the NeurIPS deadline so I would consider it a concurrent work. It may be worth citing this paper later in the camera-ready.

**Quality:**
Section 1 (Introduction) lists three precise questions explored in the paper (Line 41—44).
The experiments and theoretical explanations are well designed to concretely answer these questions.
Authors do a good job at defining minimal experimental setup and executing it thoroughly.
They begin with a common setting of training CLIP models using one/more of six existing image-text datasets, and answer each question by showing trends among a subset of trained CLIP models. In all experiments, the authors have trained multiple models when subsampling training data to capture the variance in subsampling and training runs.

**Clarity:**
The writing and presentation of this paper is excellent.
All technical details for empirical analysis are well-stated and easy to follow.
Both parts of the paper — empirical and theoretical — are broken down into sub-sections for the reader to easily find answers to each question asked in the introduction. All plots are neat and figure captions are self-contained.
Some plots often have 10 or more colors in the legend — the authors have used color-blind friendly palette and differently shaped markers to improve accessibility.
The supplementary includes some basic statistics of data used in the experiments (caption lengths, image resolution, qualitative examples, etc.)

**Significance:**
In this "scaling era", we are seeing rapid progress with large models like CLIP, ALIGN, etc. that are often trained on in-house data containing millions to billions of training instances.
The quality and concept distribution of training data plays a huge role in the success of these models.
Unfortunately due to its unprecedented scale, the training data is becoming increasingly black box. The general consensus in the field states that adding more training data will keep pushing the model performance. This paper presents some contrary evidence in the context of model robustness, which is rather not obvious. Hence, I think this is a timely contribution that will force practitioners and researchers to think more in-depth about the training data distribution when scaling CLIP-like models.

---

> ### Author Response · Authors · 2022-08-02
> **Response to Reviewer Y6Mj**
>
> Thank you for the detailed feedback! We appreciate your recognition of the strengths of the paper.
>
> Both our paper and [Data Determines Distributional Robustness in Contrastive Language Image Pre-training (CLIP)](https://arxiv.org/abs/2205.01397) highlight the importance of pre-training data in robust generalization of large image-text models, but the focus is different. [Data Determines Distributional Robustness in Contrastive Language Image Pre-training (CLIP)](https://arxiv.org/abs/2205.01397) seeks to disentangle the contribution of various factors to CLIP’s robustness, with training distribution and language supervision being the main candidates. On the other hand, our work assumes that the pre-training distribution is central to CLIP’s robustness and takes a step towards designing better pre-training datasets. Our experiments involve a strict superset of the datasets explored in this previous work (which only studies YFCC and ImageNet), and consequently are able to reveal insights into how different data sources compare in the context of various distribution shifts, as well as the implications of combining data sources on robust generalization. We will update the Background section to add the citation.
>
> We will include a reference and discussion of [Multimodal datasets: misogyny, pornography, and malignant stereotypes](https://arxiv.org/abs/2110.01963) as you suggested.
>
> We plan to make our trained models and the dataset testbed publicly available upon acceptance of our work.

---

> > ### Comment · Reviewer_Y6Mj · 2022-08-03
> > **Thank you for the response.**
> >
> > I appreciate your time to think about my suggestions, and commit to improving the final version of the paper. This discussion would make the related work more complete. I stick with my pre-rebuttal rating as I have already recommended acceptance.

---

### Official Review · Reviewer_QdN7 · 2022-07-11

**Rating:** 6
**Confidence:** 3
**Soundness:** 2 fair
**Presentation:** 3 good
**Contribution:** 2 fair

**Summary:**

This paper investigates the effects of training data distributions for vision-language pretraining. Specifically, the authors train CLIP on six publicly available datasets (image-text pairs) individually or collectively and then measure models’ robustness on ImageNet variants. In the experiments, the robustness against distribution shifts is shown as a linear trend (line) of accuracy on the base ImageNet data and its variants. Each line is compared with the y= x line (the perfect model). The empirical results suggest that 1) no single training data provides consistent out-of-distribution performance and 2) combining datasets (each has its own strength) might hurt robustness of the model. The authors also analyze the results of ensembling model outputs trained on different datasets. Lastly, the authors provide theoretical analysis on the universal line only determined by the training and two test datasets (independent from model architectures).

**Questions:**

- Have you considered other tasks (e.g., image2text/text2image retrieval)? Or, do you have preliminary results on other tasks?
- How did you decide the hyperparameters for CLIP (shown in Appendix B)?


**Limitations:**

As I mentioned in the weaknesses, the empirical results are drawn from the limited data setting, i.e., there is a significant gap between the experimental setting and the training setup of SOTA models. Also, the robustness against distribution shifts is investigated using a unimodal task (ImageNet).

**Strengths And Weaknesses:**

[Strengths]
- This paper investigates the robustness of models trained on different training data configurations. This is a nice step towards understanding the data efficiency of contrastive training.
- Overall, this paper is well-written and clearly explains the motivation, experiments, and analysis.

[Weaknesses]
My main concern is the discrepancy in scale between the experimental setting and the current SOTA models (e.g., CLIP, ALIGN, MURAL, CoCA etc.). Therefore, I’m not sure if the conclusions of this paper hold for other vision-language models trained with contrastive loss.

- Vision-language pre-training is usually done using noisy massive image-text pairs (e.g., ALIGN uses 1.8B examples). In the experiments, the size of training data (up to 15M examples) is quite different from the training setup of SOTA models.
- Although the number of negatives (i.e., batch size if using batch negatives) is an important factor of contrastive learning, it’s unclear if the authors performed an initial investigation for it (batch size of 128 sounds relatively small).
- The robustness of the models could be different in other tasks. One of the major tasks (for both pre-training and test) is image-text retrieval, but evaluation is done by a unimodal task only (ImageNet).

---

> ### Author Response · Authors · 2022-08-02
> **Response to Reviewer QdN7**
>
> We would like to thank the reviewer for the constructive comments and questions!
>
> **The size of training data is quite different from the training setup of SOTA models**: We recognize the reviewer’s concern about the scale of our experiments. However, we believe our findings also hold for models with higher accuracy for the following reasons:
> - The original CLIP paper [1] offers evidence showing that CLIP’s performance exhibits reliable linear trends across compute scales (Figure 9 of [1], which includes the ResNet50-encoder that our paper works with) and performance on natural distribution shifts (Figure 14 of [1]). The linear trends in Figure 14 of the CLIP paper [1] are analogous to the linear trends in our scatter plots.
> - In the [open-source GitHub implementation of CLIP](https://github.com/mlfoundations/open_clip) that seeks to match the performance of the original CLIP paper, the authors there also note that their models that were trained on 15M samples or fewer share the same effective robustness trend as OpenAI’s CLIP models (trained on 400M samples). Please refer to the [Why are low-accuracy CLIP models interesting?](https://github.com/mlfoundations/open_clip#why-are-low-accuracy-clip-models-interesting) remark for more details.
> - Previous work [3] has experimented with training on randomly sampled subsets of the training dataset to produce models covering a wide range of accuracies, and found that these models lie on the same effective robustness trend. This includes models trained on only 1% of the original data, which matches the scale of our experiments (5M-15M samples) relative to the scale of the original CLIP paper (400M samples).
>
> Beyond the points above indicating that our results will extrapolate to larger data scales, we would also like to point out that conducting our experiments at larger data scales would be prohibitively expensive. To provide some perspectives on the amount of resources involved in full-scale experiments:
> - One of the larger datasets that are publicly available, LAION-5B, is a substantial effort that involves 14 people actively working on it [2].
> - Training a ResNet50-encoder CLIP on the full dataset of 400M samples took 18 days on 592 V100 GPUs, according to the original paper [1].
>
> The focus of our work is to study the behavior of *a diverse set* of data sources, and performing SOTA-scale experiments on 6 datasets (and their combinations) would require an enormous amount of compute and human labor. Consequently, we rely on publicly available web-crawled datasets, the majority of those are 1-2 orders of magnitude smaller than the 400M images CLIP was originally trained on. We then make use of the effective robustness framework, which has been shown to be *scale-invariant*, to take a step towards designing smaller, but more systematic, pre-training data distribution whose generalization properties can reliably extrapolate to larger scales.
>
> **Hyperparameters for CLIP**: To clarify, we use a batch size of 128 *per GPU*, giving a total batch size of 128 * 8 = 1024 for each CLIP model trained (we trained most of our models on 8 GPUs, but some were trained on 16 GPUs). The other hyperparameters follow what [open_clip GitHub repository](https://github.com/mlfoundations/open_clip) used to match the accuracy of the original CLIP models [1] when trained on the same dataset.
>
> Previous work by Miller et al. [3] has also shown that across many evaluation settings and architectures, changes to training hyperparameters (including batch size) do *not* affect the effective robustness of the trained models.
>
> To directly answer your concern, we have additionally trained CLIP on YFCC-5M and LAION-5M subsets using a larger per-GPU batch size of 256, and found that the linear trends for these models are the same as the ones in our submission with per-GPU batch size 128. More details can be found in this figure: https://i.postimg.cc/4XT8DcZF/bs-256.png.
>
> **The robustness of the models could be different in other tasks, evaluation is done by a unimodal task only**: Our evaluation setups involve ImageNet and ImageNet-derived distribution shifts that have also been used in the original CLIP paper [1], in addition to various previous work [4, 5, 6, 7]. Per your comment, we have also run additional experiments comparing YFCC and LAION on image-retrieval tasks, with performance on MSCOCO-5K and Flickr-30K being the two axes of evaluation. In this setup, we find that our original findings still hold: CLIP models trained on a combination of inputs randomly sampled from both sources (red line) are less robust than models trained on the better data source (YFCC, blue line). More details can be found in this figure: https://i.postimg.cc/Vw7VJ5Bd/image-retrieval.png.
>
> \
> We hope our response has clarified the main concerns that you have, and we would be happy to discuss this further in the discussion phase.

---

> > ### Author Response · Authors · 2022-08-02
> > **Response to Reviewer QdN7 (cont.)**
> >
> > **References:**
> >
> > [1] Radford, Alec, et al. "Learning transferable visual models from natural language supervision." International Conference on Machine Learning. PMLR, 2021.
> >
> > [2] Laion-5b: A new era of open large-scale multi-modal datasets. https://laion.ai/laion-5b-a-new-era-of-open-large-scale-multi-modal-datasets/. Accessed: 2022-07-
> > 29.
> >
> > [3] Miller, John P., et al. "Accuracy on the line: on the strong correlation between out-of-distribution and in-distribution generalization." International Conference on Machine Learning. PMLR, 2021.
> >
> > [4] Yu, Jiahui, et al. "Coca: Contrastive captioners are image-text foundation models." arXiv preprint arXiv:2205.01917 (2022).
> >
> > [5] Wortsman, Mitchell, et al. "Robust fine-tuning of zero-shot models." Proceedings of the IEEE/CVF Conference on Computer Vision and Pattern Recognition. 2022.
> >
> > [6] Taori, Rohan, et al. "Measuring robustness to natural distribution shifts in image classification." Advances in Neural Information Processing Systems 33 (2020): 18583-18599.
> >
> > [7] Jia, Chao, et al. "Scaling up visual and vision-language representation learning with noisy text supervision." International Conference on Machine Learning. PMLR, 2021.

---

> ### Author Response · Authors · 2022-08-08
> **Follow up**
>
> Dear Reviewer QdN7,
>
> We’d like to reach out again to check if there are any additional questions/ concerns about our rebuttal that we can address before the reviewer-author discussion period ends on August 9. Thank you again for taking the time to provide constructive comments!
>
> Paper Authors

---

> ### Comment · Reviewer_QdN7 · 2022-08-09
> **Thank you for clarifying**
>
> I would appreciate the comprehensive response to my comments.
>
> Regarding the scale, I understand that the authors' claims would hold for larger models as their findings align with prior work. I agree the scope of this paper; the experimental setup is actually reasonable. Also, I would like to thank the authors for the additional analyses which is definitely a useful data point to evaluate the findings. Given these, I've adjusted my score.

---

### Official Review · Reviewer_8piD · 2022-07-11

**Rating:** 7
**Confidence:** 3
**Soundness:** 4 excellent
**Presentation:** 4 excellent
**Contribution:** 3 good

**Summary:**

This work investigates how pre-training distributions induce robustness in image-text models. They use Contrastive Language-Image Pre-training (CLIP) as a case study and evaluate it with six publicly available datasets (i.e YFCC, LAION, Conceptual Captions, WIT, RedCaps, and Shutterstock). The overall results and ablation studies show that the robustness of CLIP and other pre-trained image-text models does not depend on the quantity of the training data/data sources, but rather depends on the data quality. In addition, this work shows that mixing/combining multiple data sources/training data only dilutes the robustness of the best individual data source instead of yielding better models. Finally, the authors call for further study into dataset design, citing that web-crawling is not the effective way to build a pre-training dataset for robust generalization.

**Questions:**

Please refer to the weaknesses section.

**Limitations:**

Please refer to the weaknesses section.

**Strengths And Weaknesses:**

Strengths:
1. The paper is well written, clear, and easier to understand and follow.
2. The studied problem is very interesting and important to the NeurIPS community, given that web-based data sources/training data are widely used in pre-training large models.
3. The experimental and research methodology used clearly support the central claim of the paper.

Weaknesses:
1. This work does not provide the details on how the data sources were combined during input mixing in Section 5.1. Were the samples selected randomly or controlled? If randomly did you try selecting using different random seeds? How diverse were the selected samples? This also applies when selecting the subset of the used datasets in Section 3. It would be better to clarify this, since different samples may contribute differently during pre-training?
2. It would also be more helpful to the community if you would discuss/recommend better dataset design approaches for robust generalization. As the paper title touches on both dataset design and robustness.

---

> ### Author Response · Authors · 2022-08-02
> **Response to Reviewer 8piD**
>
> We thank the reviewer for their time and feedback!
>
> **Were the samples selected randomly or controlled:** During input mixing, samples are randomly selected from each source, and thus are representative of the data distribution of that source. To answer your question, and given the time constraint of the rebuttal, we have trained two new CLIP models on a different 5M random subset of YFCC and LAION each. We found that across the distribution shifts that we consider in our paper, using different seeds to randomly select samples does not change the linear trends. More details can be found in this figure: https://i.postimg.cc/ssW6HmFh/different-data-seeds.png.
>
> **Discuss/recommend better dataset design approaches for robust generalization:**
> * On the evaluation front, our findings indicate that building a pre-training dataset requires multiple robustness test sets, as we have shown that different pre-training datasets exhibit substantially different behaviors across distribution shifts.
> * On the dataset design front, our work suggests that each candidate data source for the overall pre-training dataset should be evaluated separately for its generalization and robustness properties. The final pre-training dataset should then contain more samples from the higher-quality pre-training sources. Creating a more "diverse" pre-training dataset by randomly sampling from all candidate sources does **not** result in a better dataset.
> * Assuming access to models separately trained on each source of interest, output ensemble is a good predictor of the robustness obtained from input mixing, and hence can significantly reduce the time to search for the right mixing strategy
> * Our theoretical analysis suggests that filtering a noisy data source with an existing robust model can improve the generalization properties of the resulting dataset.
>
> We will revise the Conclusion section of our work to summarize these takeaways.

---

> > ### Comment · Reviewer_8piD · 2022-08-05
> > **Thank you for the response**
> >
> > Thank you for taking into consideration my comments and suggestions in your final version. I stick to my decision of acceptance. Congratulations on your great work!

---

### Meta-Review · Area_Chair_eWMB · 2022-08-25

**Recommendation:** Accept
**Confidence:** Certain

**Metareview:**

This paper studies the effect of pre-training data on the robustness of pre-trained vision-language models such as CLIP. Both empirical and theoretical results are provided to show that simple scaling may not always improving robustness. This is a timely results that shed light on how to perform better pre-training to improve robustness to distribution shifts. All reviewers agree that the work is technically solid and the contribution is significant.

**Award:**

No

---

### Decision · Program_Chairs · 2022-09-14

Accept